# AKT mutant allele-specific activation dictates pharmacologic sensitivities

Tripti Shrestha Bhattarai [1,2], Tambudzai Shamu[1,2], Alexander N. Gorelick[1,2], Matthew T. Chang[1,2,7], Debyani Chakravarty [3], Elena I. Gavrila[1,2], Mark T. A. Donoghue[3], JianJong Gao[3], Swati Patel[1], Sizhi Paul Gao[1], Margaret H. Reynolds[4], Sarah M. Phillips[3], Tara Soumerai[4,8], Wassim Abida[4], David M. Hyman [4,5,6], Alison M. Schram [4], David B. Solit [1,3,4,5], Lillian M. Smyth[4,6] & Barry S. Taylor [1,2,3,5,6] ✉

AKT- a key molecular regulator of PI-3K signaling pathway, is somatically mutated in diverse solid cancer types, and aberrant AKT activation promotes altered cancer cell growth, survival, and metabolism[1–8]. The most common of *AKT* mutations (*AKT1* E17K) sensitizes affected solid tumors to AKT inhibitor therapy[7,8]. However, the pathway dependence and inhibitor sensitivity of the long tail of potentially activating mutations in AKT is poorly understood, limiting our ability to act clinically in prospectively characterized cancer patients. Here we show, through population-scale driver mutation discovery combined with functional, biological, and therapeutic studies that some but not all missense mutations activate downstream AKT effector pathways in a growth factor-independent manner and sensitize tumor cells to diverse AKT inhibitors. A distinct class of small in-frame duplications paralogous across AKT isoforms induce structural changes different than those of activating missense mutations, leading to a greater degree of membrane affinity, AKT activation, and cell proliferation as well as pathway dependence and hyper-sensitivity to ATP-competitive, but not allosteric AKT inhibitors. Assessing these mutations clinically, we conducted a phase II clinical trial testing the AKT inhibitor capivasertib (AZD5363) in patients with solid tumors harboring AKT alterations (NCT03310541). Twelve patients were enrolled, out of which six harbored *AKT1-3* non-E17K mutations. The median progression free survival (PFS) of capivasertib therapy was 84 days (95% CI 50-not reached) with an objective response rate of 25% ($n = 3$ of 12) and clinical benefit rate of 42% ($n = 5$ of 12). Collectively, our data indicate that the degree and mechanism of activation of oncogenic AKT mutants vary, thereby dictating allele-specific pharmacological sensitivities to AKT inhibition.

[1] Human Oncology and Pathogenesis Program, Memorial Sloan Kettering Cancer Center, New York, NY, USA. [2] Department of Epidemiology and Biostatistics, Memorial Sloan Kettering Cancer Center, New York, NY, USA. [3] Marie-Josee and Henry R. Kravis Center for Molecular Oncology, Memorial Sloan Kettering Cancer Center, New York, NY, USA. [4] Department of Medicine, Memorial Sloan Kettering Cancer Center, New York, NY, USA. [5] Weill Cornell Medical College, New York, NY, USA. [6] Loxo Oncology at Lilly, Stamford, CT, USA. [7] Present address: Loxo Oncology at Lilly, Stamford, CT, USA. [8] Present address: Massachusetts General Hospital, Boston, MA, USA. ✉email: taylor.lab.msk@gmail.com

The mammalian v-akt murine thymoma viral oncogene homolog (AKT) belongs to the protein kinase A, kinase G, and kinase C superfamily of serine/threonine kinases. *AKT* is a critical signaling node that translates PI3K pathway stimulation into cellular effects on cell cycle progression, survival, and metabolism. Aberrant PI3K/Akt/mTOR signaling drives many human cancers, mediated in part by mutations in *AKT*[1–3]. *AKT1* E17K is the most frequent oncogenic AKT1 mutation identified in human cancers[4–6], and has been shown to sensitize affected solid tumors to AKT inhibitor therapy[7,8]. However, there exists a long right tail of less frequent somatic *AKT1* mutations[5,6,9,10] of uncertain biological and therapeutic significance that we routinely observe in advanced cancers, but lack the information to act clinically.

In this work, we integrate computational, biochemical and experimental characterization, and an Investigator Initiated co-clinical trial framework to functionally interrogate the pathway dependence and inhibitor sensitivity of novel low incidence AKT mutations, and to expand the biomarker of sensitivity to AKT blockade in molecularly defined cancer patients.

## Results

### Identification of candidate driver mutations in AKT1, AKT2, and AKT3.
To characterize the functional consequences of AKT mutations, we defined the landscape of candidate driver mutations in all AKT isoforms. We analyzed somatic mutations in *AKT1*, *AKT2*, and *AKT3* in a cohort of 41,075 human cancers including both retrospective sequencing of largely primary and treatment-naïve tumors from The Cancer Genome Atlas cohort and various published studies combined as described previously[11], and prospective sequencing data from the tumor and matched normal specimens of 21,936 patients profiled as part of an institution-wide tumor profiling initiative[12] (see "Methods"). We characterized mutational hotspots [single-codon, in-frame insertion/deletion (indel), or those clustered in physical proximity in the cognate folded protein][9,11,13] and integrated a knowledgebase of oncogenic effects and therapeutic relevance[14].

In total, 1254 of 41,075 sequenced tumors harbored a somatic mutation in either *AKT1*, *AKT2*, or *AKT3* in the study cohort. Of these, 457 unique tumor specimens harbored a mutant allele that was considered a driver mutation by one of the three orthogonal methodologies used here. These candidate driver mutations in *AKT1-3* were most common in breast cancers (4.3% of cases, excluding presumed passenger mutations), an overall population frequency consistent with reports from prior studies of more limited datasets[4,15–17]. Overall, AKT mutations were most common in hormone-driven cancers, albeit a long right tail of less frequent incidence existed across many solid tumor types (Fig. 1a). As expected, *AKT1* E17K was the most common, and ~22-fold more prevalent than the next most common AKT mutation- the *AKT1* L52 hotspot (Fig. 1b). Other *AKT1*, *AKT2*, and *AKT3* candidate driver mutations comprised a long tail of uncommon mutations, but together accounted for 28% of all patients with a candidate AKT driver mutation (Fig. 1c and Supplementary Table 1).

The majority of hotspot mutations affected the PH domain, consistent with its role in membrane engagement, translocation, and protein activation. Although not common, *AKT2* and *AKT3* mutations also arose in residues paralogous to hotspots in *AKT1*. These mutations included E17K as well as *AKT2* D324 and *AKT3* D320, which are paralogous to *AKT1* D323, the only kinase domain single-codon hotspot identified. We also identified a cluster of more complex in-frame paralogous insertion and duplication mutants in *AKT1* and *AKT2* (Fig. 1d). Finally, we identified a cluster of physically adjacent mutations evident only

when the cognate protein is folded in three dimensions (3D)[13]. Eight mutations defined this cluster, which lies at the PH-Kinase domain interface. This 3D cluster of mutations included physically abutting hotspots *AKT1* E17 and D323 as well as private or rare non-hotspot mutant residues R15, R23, R25, V320, L321, and E322, all of which lie within 5 angstroms of one another (Fig. 1e).

### AKT1/2 candidate driver mutations activate PI3K pathway.
To assess the ability of candidate AKT driver mutations to induce aberrant PI3K pathway activity, we stably expressed the wild-type (WT) allele and 25 unique missense and small in-frame indel mutations in *AKT1* and *AKT2* in MCF10a primary breast epithelial cells, the lineage with the greatest frequency of *AKT* mutations in cancer (Fig. 1a). These studies encompass 88% of all human tumors harboring the aforementioned candidate *AKT1-3* driver mutations from nearly all affected cancer types (Supplementary Fig. 1a). Compared to parental cells or those expressing WT *AKT1*, expression of seven unique *AKT1* missense mutants induced elevated levels of phosphorylated AKT (p-AKT) and downstream substrates p-GSK3α/β, p-PRAS40, and p-S6 when assessed in assay media lacking growth factors (see "Methods"; Fig. 2a). This included all those alleles identified as statistically significant hotspot mutations. We also expressed four *AKT1* mutations that were neither statistically significant hotspots nor remarkable for their paralogy across AKT isoforms (E267G, E341K, R370C, and E464K). Indeed, none of these mutant alleles induced expression of p-AKT, including the E267G and R370C mutations that were previously shown to cause tumor formation when expressed in HA1E-M cells[18]—a difference with prior work that may reflect a lineage-specific difference in their ability to stimulate pathway activity or perhaps was attributable to HA1E-M cells expressing constitutively active MEK1[DD]. Finally, AKT1 R15Q did not induce AKT activation greater than WT AKT1 despite its clustering in physical proximity to other oncogenic *AKT1* hotspot mutations (Fig. 1e), consistent with prior data indicating that it does not participate in phosphatidylinositol binding[19]. Expression of the four paralogous mutant residues (W22R, D44N, R48H, F55Y) mostly failed to induce elevated p-AKT expression. While we failed to detect phosphorylated AKT by immunoblot in MCF10a cells expressing *AKT1* F55Y, this mutation nevertheless induced both p-S6 and p-PRAS40 like other driver mutants and was therefore considered activating for the purposes of these analyses (Fig. 2a). Overall, consistent with previous studies, the expression of several *AKT1* non-E17K missense mutations resulted in activation of downstream AKT effectors in the absence of growth factors, the majority of which lie in close proximity to the PHD-KD interface and form crucial interactions that regulate protein activation (see "Methods").

Our computational analysis also identified a cluster of novel in-frame indel mutations in a paralogous region of the PH domain of *AKT1* and *AKT2* proximal to the two known hotspots L52 and Q79 (Fig. 1d and Supplementary Fig. 1b). We thus generated stable MCF10a cells expressing eight of these indels [*AKT1* T65-I75dup, E66-Q79dup, and P68-C77dup; *AKT2* C60-I75dup, R67-L78dup, P68-W80dup, I75-I84dup, and L78-Q79ins(HANTFVIRCL)] and assessed their ability to induce growth factor-independent activation of AKT signaling. Unlike the variable pathway activation of missense hotspots, all of the in-frame indels tested led to hyper-phosphorylation of AKT1 (T308 and S473) or AKT2 (T309 and S474), and robust growth factor-independent activation of the pathway (Fig. 2b, c). While expressed at lower levels, the indels induced p-AKT levels that significantly exceeded those of the well-established oncogenic E17K mutation and induced a concomitantly higher level of p-S6 and p-PRAS40.

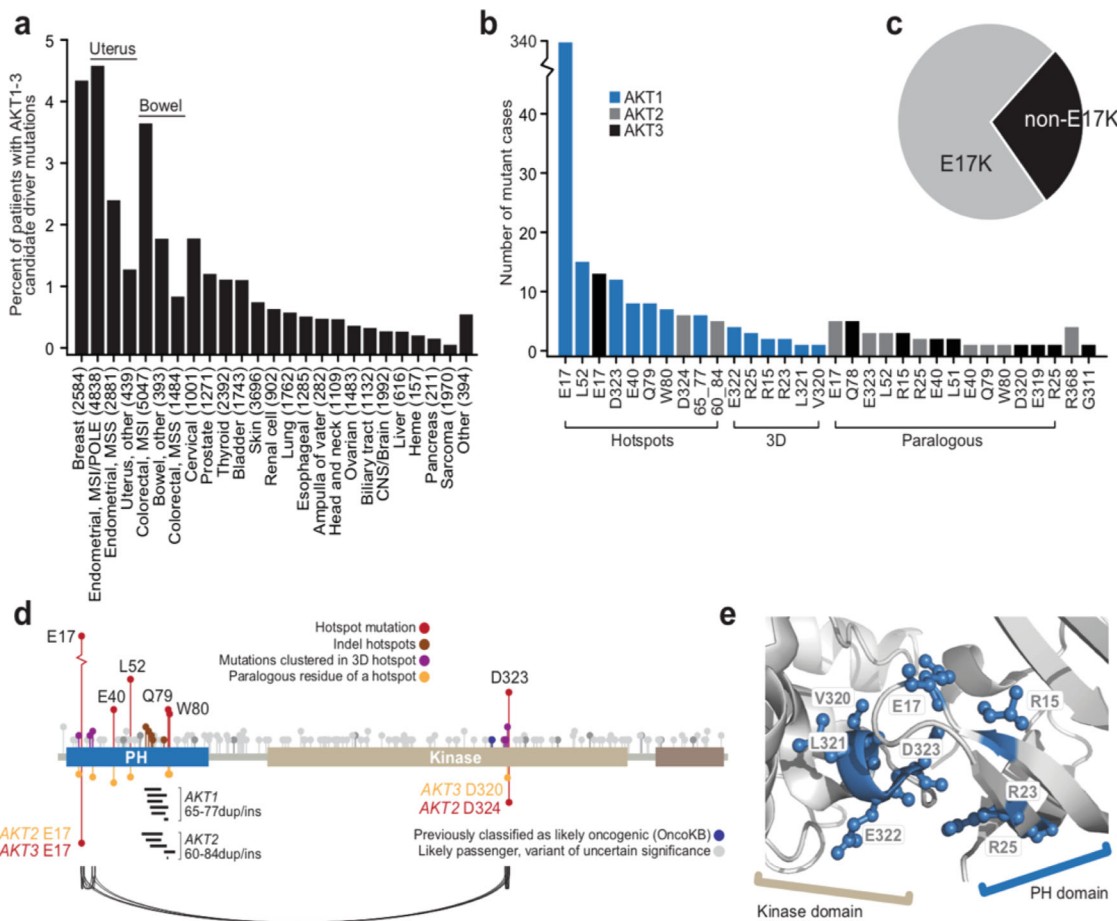

**Fig. 1 Somatic mutations in AKT in human cancers. a** The frequency of known and candidate driver mutations in *AKT1*, *AKT2*, and *AKT3* in diverse primary and metastatic human cancers as determined from population-scale sequencing of 41,075 patients (frequencies based on only known or candidate driver mutations only, excludes presumptive passenger mutations). **b** The number of affected samples for each of the individual mutations in *AKT1*, *AKT2*, and *AKT3* indicates a long tail of increasingly uncommon candidate driver mutations. **c** The fraction of cases harboring candidate driver mutations in *AKT1-3* that are E17K or not. **d** Schematic representation of the domain structure of AKT1 and the position of different classes of mutations (see inset legend) identified as candidate driver alterations including those in paralogous residues in *AKT2* and *AKT3* (orange). Arcing lines reflect physically proximity in the cognate folded protein (panel d). Small in-frame indels are shown as horizontal lines and target a paralogous cluster in *AKT1* and *AKT2*. **e** A cluster of physically adjacent mutations (blue) in the AKT1 protein structure within 5 angstroms of each other and two single-codon hotspots (E17 and D323). The PH and Kinase domains are labeled as in panel (**d**).

**Phenotypic consequences of activating AKT1/2 mutations.** To assess the relative oncogenic potential of AKT indels, we stably expressed AKT1 and AKT2 missense and indel mutations in IL-3-dependent murine pro-B Ba/F3 cells. Consistent with our results in MCF10a cells, Ba/F3 cells expressing the AKT1/2 indel mutants potently induced p-AKT and downstream targets (Fig. 2d), which was associated with IL-3-independent cell proliferation that was several-fold greater than the effect associated with E17K and other activating missense mutations (Fig. 2e). The superior expression of some mutant alleles in Ba/F3 cells clarified their affect, such as F55Y which induced p-AKT, p-S6, and p-PRAS40 despite ambiguous results at lower expression in MCF10a cells. Collectively, these results suggest that these newly discovered and recurrent *AKT1* and *AKT2* in-frame insertion and duplication mutants uniformly hyper-activate AKT signaling and are associated with more profound mutant AKT phenotypes than those associated with more common missense AKT hotspot mutations (Supplementary Fig. 2).

**Structural impact of AKT1/2 indel mutations.** We next sought to define the mechanistic basis for the higher degree of p-AKT induced by the duplication mutants compared to missense drivers. We performed molecular dynamic (MD) simulation analysis leveraging an established structure of AKT1 in closed conformation that includes both PH and kinase domains (see "Methods"). This analysis indicated that the canonical E17K mutation only modestly disrupts the auto-inhibitory interactions between the PH and kinase domains in the AKT1 structure as measured by predicted changes in inter-residue distance between these domains, consistent with expectation[4,10,19–23]. By contrast, variable length paralogous in-frame indel mutations typified by P68-C77dup induced systematic physical displacement throughout the protein structure, effectively forcing a more extended PH-out conformation (Fig. 3a, b). Among the regions of greatest displacement in position and spatial orientation was the T308 residue (Fig. 3c), which changed from a buried position in WT and E17K to become partially exposed on the P68-C77dup protein surface (relative solvent accessibility: from 10–20% to 50–60% exposed in P68-C77dup) and therefore more accessible for PDK1-mediated phosphorylation. This structural deformation is consistent with the hyper-phosphorylation of T308 we observed specifically in the AKT indels compared to the AKT missense activating mutants (Fig. 2b, c). Relative to E17K, the P68-C77dup mutant also re-localized Q79, breaking its stabilizing

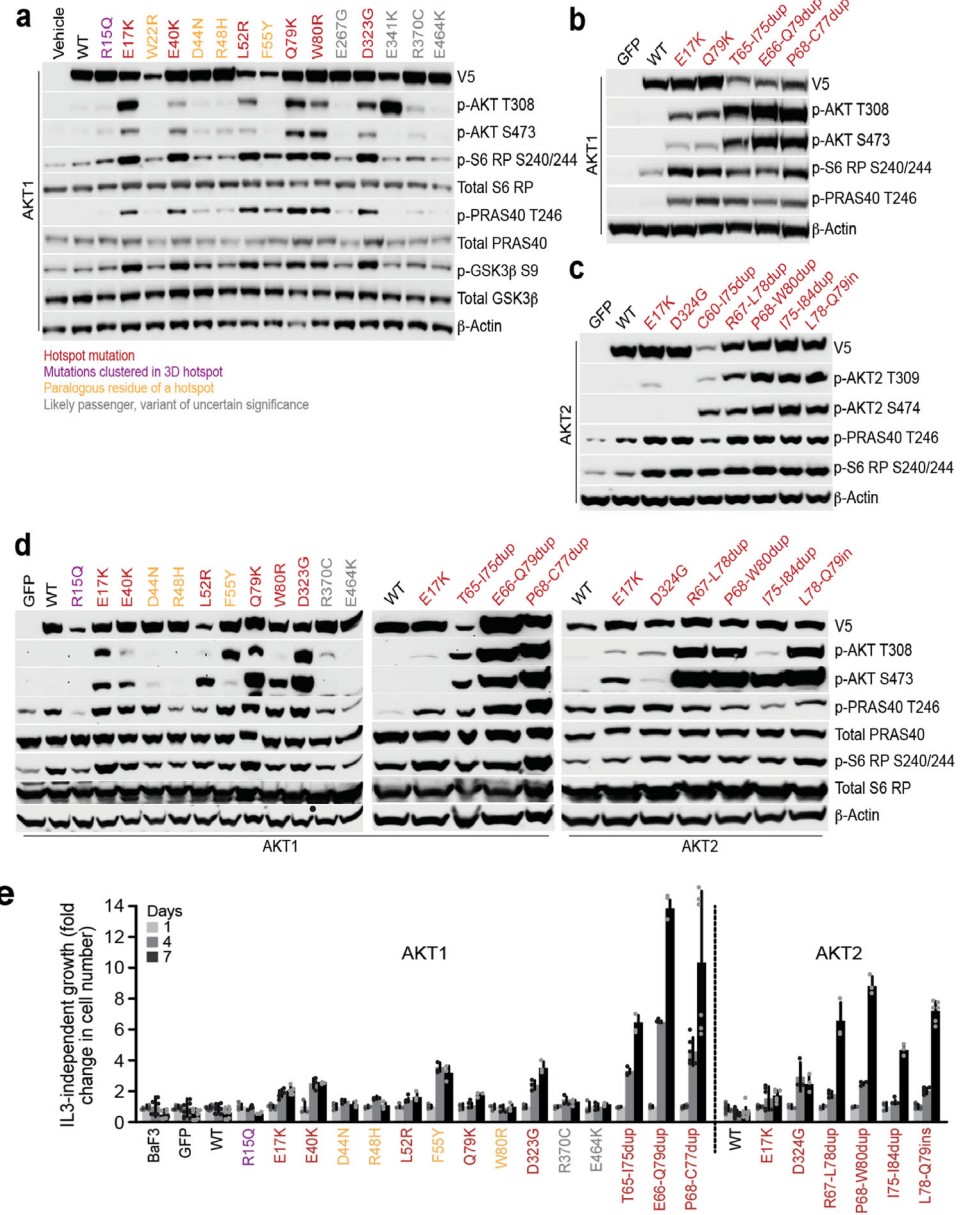

**Fig. 2 Diverse AKT alleles hyper-activate PI3K signaling to differing degrees. a** MCF10a cells stably expressing the indicated *AKT1* substitutions were incubated in assay medium overnight. Expression and phosphorylation levels were assayed by Western blot, indicating that all mutations identified as hotspots induced activation of AKT and downstream targets, but few other paralogous residues or those identified by protein structure analysis were similarly activating. **b** As in panel (**a**) but for a series of small in-frame duplications in AKT1, showing that while E17K and Q79K are activating, AKT1 duplications induce far higher levels of phosphorylated AKT and robust pathway activation. **c** As in panels (**a**, **b**) but showing the effect of *AKT2* activating mutations including single-codon *AKT2* hotspots (E17K and D324G) and *AKT2* indels. **d** WT, E17K, and various additional indicated missense and indel mutants in AKT1 and AKT2 were stably expressed in murine pro-B Ba/F3 cells and assessed for their ability to induce phosphorylation of AKT and downstream targets in the absence of interleukin-3 (IL-3) by western blot analysis of whole cell lysates. Results shown are representative images from experiments performed at least three times with multiple batches of stable cells. **e** Mutants as in panel d, assessed for their ability to promote IL3-independent Ba/F3 cell proliferation. Error bars are standard deviations from the mean. Results were derived from at least two independent experiments with triplicates for each experiment. Source data are provided as a Source Data file.

hydrogen bond with N53, further facilitating PH domain binding to inositol and thereby membrane localization[19,22] (Fig. 3d). Repeating this analysis using an AKT1 structure that is ATP analog-bound and therefore represents its open conformation[23], we found that P68-C77dup had a more modest structural effect than it did on AKT1 in closed conformation (Supplementary Fig. 3a). Moreover, while E17K did not result in a significant difference in the distance between strongly interacting key residues between the PH and kinase domains in either the closed or

open conformation compared to the WT, P68-C77dup resulted in a significant increase in these distances in the closed conformation, an effect that was weaker when assessed in an open conformation where these interacting residues had large initial distances (Supplementary Fig. 3b). These results are consistent with the duplication mutant promoting an open-like conformation. Collectively, these simulations indicated that indels typified by P68-C77dup drive structural displacement that alters the spatial orientation of functionally critical residues primarily in the

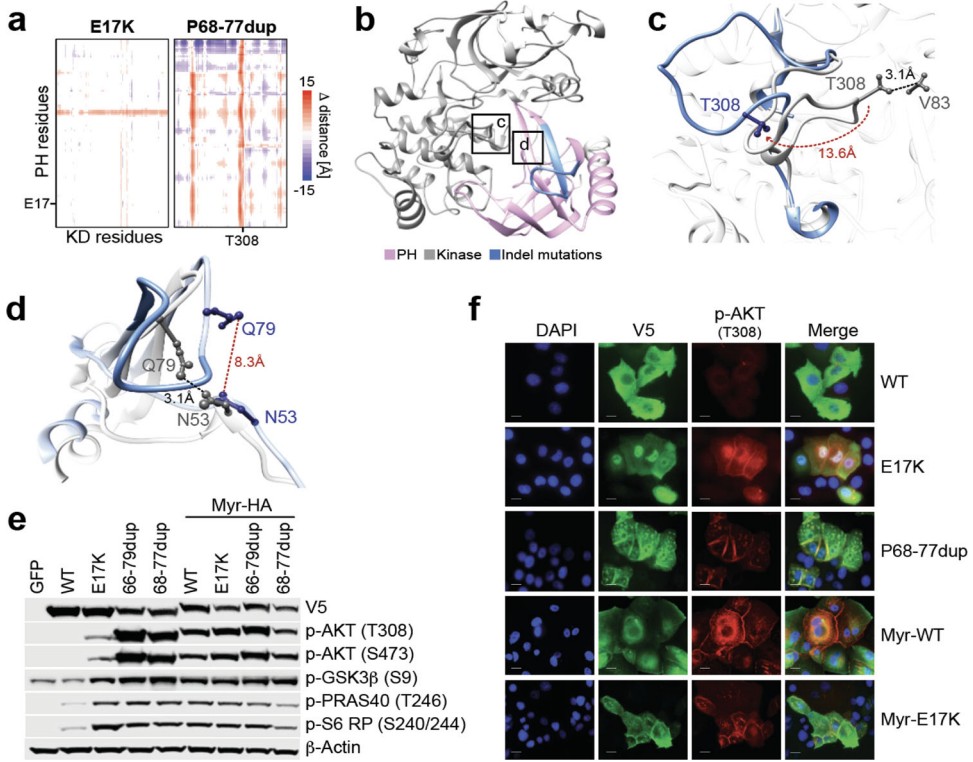

**Fig. 3 Structural and signaling impact of AKT in-frame indels. a** Molecular dynamic simulation of AKT1 indicated systematic displacement of the structure by the P68-C77dup mutation as compared to the E17K hotspot, as measured by changes in inter-residue distance between the PH and kinase domains. Duplication mutant-specific displacement targeted the key regulatory phosphorylation site T308, among others. **b** The structure of AKT1 indicating the region affected by the cluster of paralogous in-frame indels in *AKT1* and *AKT2* (light blue) in the PH domain (purple). The kinase domain (KD) is shown in light gray and boxed are the two regions of greater detail shown in the indicated panels. **c** A detailed view of the 13.6 angstrom T308 displacement by the P68-C77dup mutation. **d** A detailed view of N53 in the inositol-binding region of the PH domain that is displaced by the duplication mutant breaking its WT hydrogen bonding with Q79. **e** MCF10a cells stably expressing either GFP, WT AKT1, or the indicated missense or duplication mutants and their myristoylated counterparts were incubated in assay medium overnight, and whole-cell lysates were prepared and immunoblotted with the indicated antibodies in order to assess pathway activation. Source data are provided as a Source Data file. **f** MCF10a cells stably expressing WT or mutant AKT1 and their myristoylated counterparts were incubated with assay media for four hours, after which the cells were fixed, permeabilized, probed with antibodies for p-AKT (T308) and V5 tag (for AKT expression) and examined under fluorescence microscope in order to determine the intracellular localization of the indicated AKT1 constructs. For panels (**e**–**f**), results are representative of at least three independent experiments performed. Scale bar: 40 μm.

PH domain, thereby decreasing the structural stability of the PH and kinase domains overall and abolishing stabilizing bonds that promote the closed conformation of WT AKT[22].

Our MD simulations suggest that the higher p-AKT levels attributable to AKT indels are due to their constitutively open conformation compared to the closed or partially open conformations of WT and E17K. Binding to membrane lipids is associated with open AKT conformation and constitutive activity[21]. We therefore reasoned that adding a myristoyl tag (hereafter Myr-AKT), thereby targeting AKT to the plasma membrane[24], would induce an open conformation and increased activation for WT and E17K, but produce little additional change for already saturated AKT indels. We therefore stably expressed in MCF10a cells AKT1 E17K, P68-C77dup, and E66-Q79dup as well as their myristoylated forms. Both myristoylated forms of WT and E17K (Myr-AKT1-WT and Myr-AKT1-E17K, respectively) induced levels of p-AKT greater than their non-myristoylated counterparts (Fig. 3e). Immunofluorescence (IF) microscopy revealed that, while WT AKT1 was dispersed throughout the cytoplasm, most of the p-Akt (T308) in AKT1 E17K cells was either peri-nuclear in localization or membrane-associated (Fig. 3f). On the other hand, Myr-AKT1-WT and Myr-AKT1-E17K were

predominantly membrane-localized, and as expected, demonstrated strong induction of p-Akt (T308) signal over their non-myristoylated versions. By contrast, Myr-AKT1-E66-Q79dup and Myr-AKT1-P68-C77dup induced no greater p-AKT, p-PRAS40, or p-S6 than did the non-myristoylated indels. Indeed, unmyristoylated AKT1 P68-C77dup showed a distinct pattern of constitutive membrane localization and AKT phosphorylation that resembled myristoylated forms of AKT, indicative of robust kinase activation (Fig. 3e–f). A similar pattern was evident for indels in *AKT2* (Supplementary Fig. 4). Moreover, by inhibiting mutant cells with either the covalent non-specific PI3K inhibitor Wortmannin or the PIK3CA inhibitor BYL-719, we also found that the membrane-binding activity and subsequent activation by the indel (but not missense) mutants was independent of PIP3 formation (Supplementary Fig. 5). Overall, these data suggest that *AKT1/2* indels are a unique class of oncogenic mutations whose distinct structural consequences cause the protein to adopt a more open PH-out conformation, thereby promoting membrane engagement, inducing AKT hyperphosphorylation and ATP-binding that cooperatively protects AKT from de-phosphorylation[25], and triggering pathway activation far greater than any missense mutation identified to date.

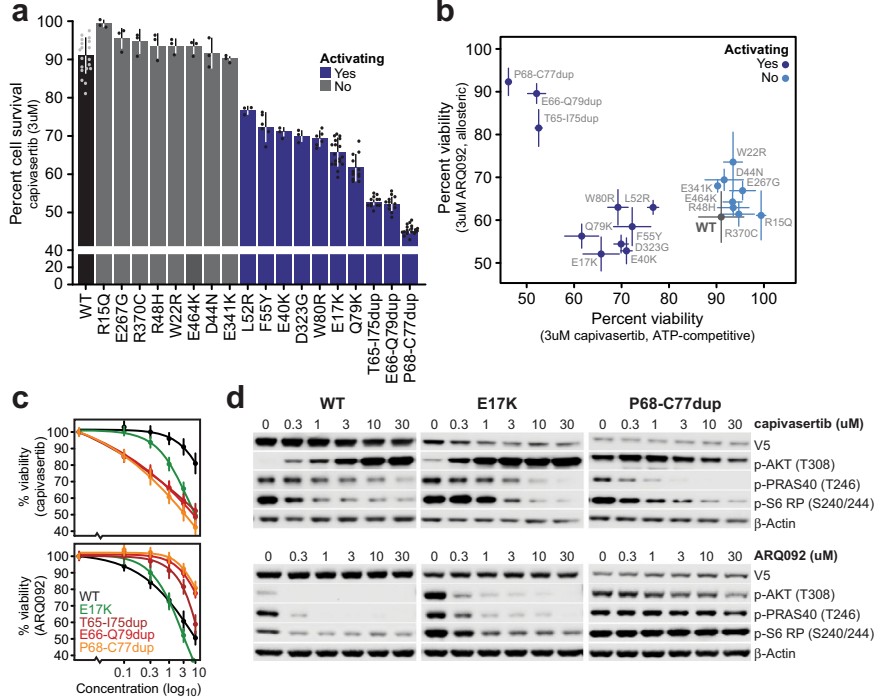

**Fig. 4 Pre-clinical sensitivity of diverse AKT mutations to AKT inhibition. a** The viability of MCF10a cells stably expressing either WT or the indicated mutations in AKT1 was assessed after treatment with 3 μM of the ATP-competitive AKT inhibitor capivasertib. Legend indicates which of the alleles are activating. **b** As in panel A, cells expressing the indicated mutants were assessed for viability after treatment with either capivasertib or the allosteric inhibitor ARQ092. Points represent mean viability percentage and error bars indicate the 95% confidence intervals. For panels (a-b), mean viability was determined from at least two independent experiments. **c** MCF10a cells stably expressing wild-type AKT1, the known oncogenic E17K, or indel mutants were treated with indicated concentrations of capivasertib or ARQ092 (top and bottom, respectively), and cell viability was assessed 72 hours post-treatment. Results derived from 18 values from six independent experiments. **d** Whole-cell lysates from MCF10a cells expressing AKT1 WT, E17K, and P68-C77dup were harvested four hours post-treatment with either capivasertib (top) or ARQ092 (bottom), and pathway inhibition was assessed by western blot. Results are representative images from at least two independent experiments. Error bars in panels (**a**) and (**c**) represent standard deviations from the mean. Source data are provided as a Source Data file.

**Differential pharmacologic sensitivities of AKT mutants.** As indel mutations are structurally distinct from missense mutations in AKT, and lead to different degrees of activation and membrane affinity, we sought to test differences in their drug sensitivity. Such findings would impact the clinical management of cancer patients harboring such mutations, in line with our previous finding that selective AKT inhibition has single-agent activity in a histology-agnostic basket study of AKT1 E17K-mutant solid tumors[7]. Using the same agent, the ATP-competitive kinase domain inhibitor capivasertib (AZD5363), we assessed whether such activating *AKT1/2* non-E17K mutations similarly sensitize cells. Cell viability assays demonstrated that all activating AKT1 missense mutations (Fig. 2a) sensitized cells to capivasertib treatment, to varying degrees, as indicated by significant growth inhibition (Fig. 4a). By contrast, treatment with capivasertib in MCF10a cells expressing WT or non-activating mutants of *AKT1* did not significantly alter cell survival. *AKT1* indels were significantly more sensitive to capivasertib at all drug concentrations (Fig. 4b, c), possibly due to greater ATP-competitive inhibitor binding[22] to the binding pocket of the more open conformation of the kinase. This result is consistent with the more potent induction of PI3K pathway activation by the indels. Capivasertib treatment in AKT1 WT, E17K-mutant, and indel-mutant cells caused a dose-dependent decrease in the phosphorylation of downstream targets (p-PRAS40 and p-S6 RP). However, the paradoxical AKT1 hyperphosphorylation[26,27] observed in AKT1 WT and E17K-mutant cells was not as prominent in P68-C77dup-mutant cells, perhaps because the phosphorylation levels

in these cells are near saturation even in the absence of capivasertib treatment or other forms of stimulation (Fig. 4d). Overall, the level to which AKT was activated by driver mutations with distinct structural consequences on the cognate protein correlated with sensitivity to ATP-competitive inhibitor treatment.

In contrast to ATP-competitive AKT inhibition, we observed compound and mutant-specific sensitivity of AKT-mutant cells to two different allosteric AKT inhibitors (MK2206 and ARQ092). Cells expressing the activating W80 mutation, which is a critical amino acid for allosteric inhibitor binding, were insensitive to MK2206[20,27,28] but had a slightly improved response to ARQ092 (Fig. 4b and Supplementary Fig. 6). This difference likely resulted from the different binding dynamics, selectivity, distinct mechanism of AKT inhibition, and greater potency of ARQ092[29]. Unlike the variable sensitivities of missense-mutant cells to both ATP-competitive and allosteric AKT inhibitors, cells expressing diverse *AKT1* and *AKT2* indel mutants [*AKT1* P68-C77dup, E66-Q79dup, and T65-I75dup; *AKT2* P68-W80dup, I75-I84dup, L78-Q79ins(HANTFVIRCL)] were resistant to treatment with allosteric inhibition, in a mechanism-of-activation rather than compound-specific manner, as resistance among the indel-expressing cells was evident for both ARQ092 and MK2206 (Fig. 4b, d and Supplementary Figs. 6 and 7). IF microscopy data revealed that, while cells expressing WT AKT1 and the E17K mutant showed significant diminution of membrane-bound p-AKT signal upon treatment with allosteric AKT inhibitors (indicative of AKT inactivation), both membrane localization and the intensity of p-AKT remained intact in cells expressing AKT1

P68-C77dup, further corroborating the lack of effectiveness of allosteric inhibitors in indel-mutant cells (Supplementary Fig. 8). Overall, the sensitivity of cells driven by distinct AKT mutations to allosteric AKT inhibitors did not correlate with pathway addiction but appeared driven instead by the binding dynamics of allosteric agents at the PH-kinase domain interface. Allosteric AKT inhibitors bind and stabilize the PH-in or closed conformation of the AKT enzyme, which abrogates the regulatory site phosphorylation and membrane association of both WT and E17K AKT (as shown in Supplementary Figs. 6 and 8). However, the structural displacement induced by the *AKT1/2* indels produce a conformational change that renders affected cells refractory to allosteric inhibition ($IC_{50}$ data for all inhibitors and mutants, Supplementary Table 2). To model this mechanism of indel mutant insensitivity to allosteric inhibitors, we performed thermal shift assays that expose proteins to increasing heat to determine their melting temperature, with unstable structures denaturing and aggregating at lower temperatures. We tested AKT1 WT, E17K, and multiple indel mutations. Unlike WT or E17K-mutant AKT1, proteins harboring the indels readily denatured at lower temperatures, likely due to their open conformation and loss of stabilizing inter-domain interactions as suggested by our MD simulation studies (Supplementary Fig. 9). Collectively, these results suggest that the differential sensitivities of individual AKT indel mutants to these two classes of AKT inhibitors are specific to the distinct mechanisms of action of the inhibitors and that increased sensitivity to ATP-competitive inhibition comes at the cost of allosteric inhibitor resistance.

**Rare activating AKT variants are clinically targettable**. Guided by these computational-experimental results, we opened an investigator-initiated tissue-agnostic clinical trial of patients harboring *AKT1-3* alterations (ClinicalTrials.gov NCT03310541) that excluded ER+ breast cancer patients harboring *AKT1* E17K mutations studied previously (NCT01226316)[30]. AKT inhibitors of diverse mechanisms of action are being tested clinically at present. However, based on our data indicating that AKT indel mutant cells were resistant to allosteric inhibition, we chose to test the efficacy of capivasertib in this trial. We enrolled a total of twelve patients across eight different cancer types, including six patients with non-*AKT1* E17K mutations (*AKT1* D323G, E40K, L52R; *AKT2* E17K, L78-Q79ins(HANTFVIRCL); *AKT3* E17K) in this signal-seeking pilot study (Supplementary Fig. 10a and Supplementary Table 3). Given the rarity of this patient population, the clinical trial was closed early due to slow accrual. Observed toxicity was consistent with the known side effect profile of capivasertib, with diarrhea and nausea as the most common treatment-emergent adverse events (Supplementary Table 4)[7]. Two patients required dose reductions, one for grade 3 acute kidney injury and the other for grade 2 diarrhea and nausea. No patient discontinued therapy due to toxicity. The objective response rate (ORR) was 25% ($n = 3$ of 12) and the clinical benefit rate was 42% ($n = 5$ of 12; median PFS of 84 days, 95% CI 50-not reached; Supplementary Fig. 10b, c). Of the six patients with non-*AKT1* E17K mutations, two patients had prolonged partial responses, one had stable disease lasting 16 weeks, and three had progressive disease (Fig. 5a). An *AKT1* L52R-mutant endometrial cancer patient had a durable partial response (PR) lasting nearly a year (Fig. 5b). A heavily pre-treated *AKT1* D323G-mutant breast cancer patient with ER-positive HER2-negative mixed ductal and lobular carcinoma had stabilization of her disease lasting 16 weeks before progressing. Among the first to enroll was a patient with metastatic prostate cancer that previously progressed on enzalutamide therapy, and whose pre-

enzalutamide tumor harbored an in-frame *AKT2* L78_Q79ins(HANTFVIRCL), an allele that we assessed in vitro for biochemical activity, transformation potential, and therapeutic sensitivity (Fig. 2c–e and Supplementary Fig. 7). Following initiation of capivasertib treatment, the patient had a rapid PR (38% reduction in target lesions). Concomitantly, the patient had a decrease in PSA (from 21 to 7 following four weeks of treatment) and marked improvement in disease-related pain (Fig. 5c). This patient ultimately remained on capivasertib therapy for 13 months before eventually progressing, which is a length of therapy that exceeded what was previously achieved with enzalutamide. Upon sequencing a biopsy obtained from the post-progression tumor, we identified the acquisition of a focal amplification of *IRS2* that was absent from the AKT inhibitor-naïve tumor at trial enrollment. Prior reports indicate that AKT inhibition and subsequent loss of mTORC1-mediated feedback can lead to upregulation of IRS1/2[3,31,32], thereby promoting potent re-activation of both Ras/ERK and PI3K signaling pathways[33,34]. Consequently, amplification of *IRS2* may serve as a mechanism of acquired resistance to AKT inhibitor therapy in this patient (Fig. 5d). Overall, these results are consistent with the hyperactivation of AKT signaling induced by indel mutations and their hypersensitivity to AKT inhibition we observed in vitro, and suggest that activating non-E17K mutations in *AKT1-3*, including in-frame indels, may broaden the sensitizing biomarker of AKT inhibition, albeit in a mechanism-of-action-specific manner.

## Discussion

Our genomic, computational, experimental, and clinical data indicate that a long right tail of activating non-E17K mutations in *AKT1* and *AKT2* induce aberrant PI3K signaling to varying degrees via distinct structural deformations of oncogenic AKT that dictate distinct pharmacologic sensitivities to AKT inhibitors of different mechanisms of action, thereby broadening the predictive biomarker of AKT inhibitor treatment in advanced cancers. Lack of sufficient knowledge about the vast majority of mutations identified prospectively in cancer patients severely limits the clinical applicability of precision medicine in oncology. Our data ultimately seek to address this key hurdle by characterizing rare AKT mutations, thus facilitating the clinical matching of such molecularly defined patients to suitable targeted therapy. Nevertheless, therapeutic vulnerabilities to AKT inhibition seem to be mutant allele-specific and will require mechanism-of-action-specific pharmacologic targeting, similar to the allele-specific differences emerging in other cancer genes[35,36]. Moreover, occult rare driver mutations not identified by the present analysis may still emerge over time with new patient populations, increasing sample sizes, new computational methods, or orthogonal modalities of screening and characterization. Moreover, mechanisms of AKT activation beyond somatic mutations including rare focal amplifications and fusions will be critical to routinely profile and understand functionally, along with the co-mutational context of AKT-mutant tumors that modify tumor dependence on this target. Ultimately, similar co-clinical trial frameworks such as we leverage here will dissect the mutant allele-specific differences in pathway activation that necessitate distinct therapeutic strategies to improve the outcomes of molecularly defined cancer patients.

## Methods

**Mutant allele discovery**. We assembled somatic mutational data from 41,075 retrospectively and prospectively sequenced human cancers. All mutational data from the studies associated with The Cancer Genome Atlas were derived from the single multi-center mutation calling in multiple cancers project (MC3)[37], and were combined with retrospective sequencing data from various published studies using an approach we have described previously[11], To these data we added somatic mutational data from 21,936 active cancer patients profiled with MSK-IMPACT

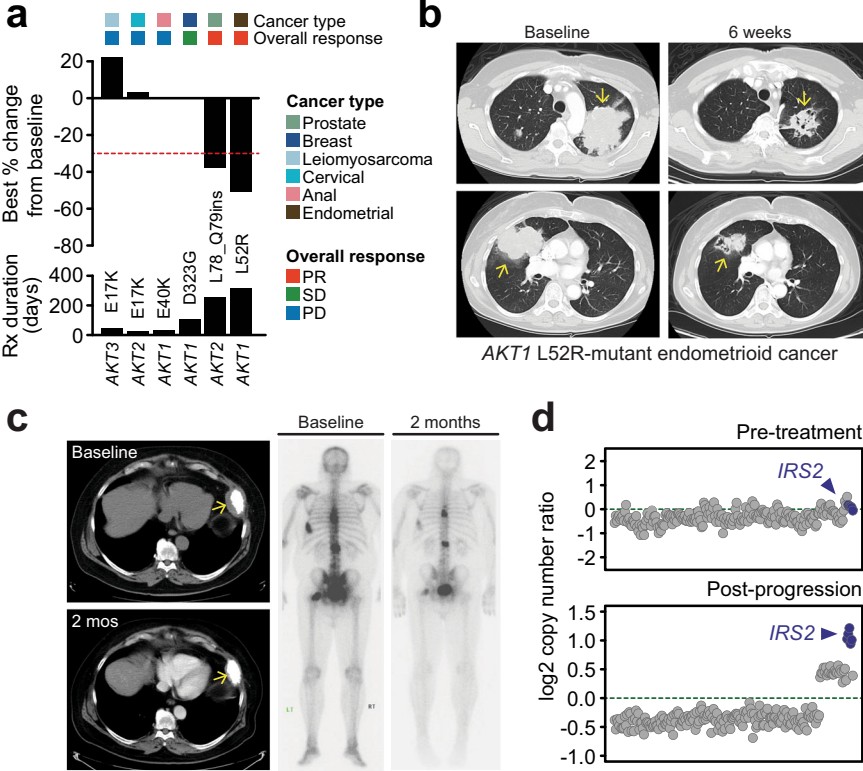

**Fig. 5 Clinical sensitivity of diverse AKT mutations in a phase II basket trial of capivasertib. a** The efficacy of ATP-competitive AKT inhibition was explored in a phase II clinical trial of capivasertib in patients harboring *AKT1-3* alterations (excluding *AKT1* E17K-mutated ER+ breast cancer). The clinical response of patients to capivasertib therapy with tumors harboring non-E17K *AKT1-3* mutations is shown. **b** Computed tomography (CT) indicating a partial response (indicated by arrows) in an endometrial cancer patient with an AKT1 L52R mutation that lasted nearly a year. **c** Response of a castration-resistant metastatic prostate cancer patient to capivasertib treatment whose tumor harbored a novel *AKT2* L78_Q79ins(HANTFVIRCL) indel mutation. Left and right are CT and bone scans at baseline and two months after treatment initiation. The left panel shows the resolution of the soft tissue metastatic compartment (yellow arrow). **d** Sequencing of pre-treatment and post-progression tumor tissue indicated the acquisition of a focal amplification of *IRS2* (blue; *x*-axis is chromosome 13) present only in the tumor after AKT inhibitor resistance.

targeted sequencing panel as part of our institutionally prospective sequencing initiative[12,38]. Among these prospectively sequenced cancer patients, 45.2% and 54.8% were male and female, respectively, with an age at the time of analysis of 62 ± 15 years (mean and standard deviation, interquartile range is 18). After excluding studies and samples of insufficient quality and breadth, we performed hotspot analysis of *AKT1, AKT2,* and *AKT3* as previously described[9,11,13]. *AKT1-3* mutational hotspots were considered significant if they exceeded a false discovery rate of 1% (*q* value < 0.01).

**Cell lines and culture conditions**. MCF10a cells were generously shared by the laboratory of D.B. Solit and maintained in DMEM/F-12 base medium containing 5% horse serum and other supplements (20 ng/ml EGF, 0.5 mg/ml hydrocortisone, 100 ng/ml cholera toxin and 10μg/ml insulin) (complete growth medium). 293-FT cells were obtained from ATCC and maintained on DMEM supplemented with 10% FBS and 2 mM glutamine. Ba/F3 cells were kindly provided by the laboratory of R. Levine and cultured in RPMI + 10% FBS + 10 ng/ml IL-3 (growth medium). For experimental studies, an assay medium was utilized in which growth factors were withdrawn. For MCF10a cell experiments, assay medium was DMEM/F-12 base medium containing 2% horse serum, hydrocortisone, and cholera toxin. For Ba/F3 cell experiments, assay medium was only RPMI plus 10% FBS without IL3.

**Plasmids, cloning, and stable line generation**. AKT1 WT in the donor vector pDONR223 was kindly provided by laboratory of J. Baselga. pcDNA3 AKT2 WT (plasmid #16000) and pcDNA3 Myr HA AKT1 (plasmid # 9008) plasmids were acquired from Addgene. Myr HA AKT1 and AKT2 (after removal of the HA tag) plasmids were then cloned into the entry vector pENTR/D-TOPO (ThermoFisher Scientific #K240020) according to manufacturer's instructions. All AKT substitution and indel mutants were generated by site-directed mutagenesis using either KAPA HiFi polymerase (KAPA Biosystems), or Q5 mutagenesis kit (New England Biolabs), and verified by Sanger sequencing. Verified constructs were ultimately sub-cloned into gateway lentiviral vector pLX302 using LR Clonase II enzyme mix (Catalog # 11791020, ThermoFisher Scientific). Lentiviruses encoding WT or mutant AKT1/2 constructs were packaged in 293FT cells; the supernatant media

containing viral particles was filtered through 0.45μm filters and used to infect MCF10a and BA/F3 cells. Cells expressing the lentiviral constructs were selected with Puromycin (2.5 μg/ml) for stable line generation.

**Western blotting and antibodies**. MCF10a or Ba/F3 cells stably expressing WT and mutant AKT were seeded in either 6-well plates or 60 mm dishes, and after overnight exposure to the assay medium, the cells were lysed, sonicated and 30 μg protein was loaded onto SDS-PAGE gels, transferred to nitrocellulose membranes, and immunoblotted for p-Akt and other downstream molecular targets of PI3K pathway activation. Antibodies for p-Akt (T308) (D25E6) [dilution 1:1000], p-Akt (S473) (D7F10 and D9E) [dilution 1:1000], p-PRAS40 (T246) [dilution 1:1000], p-S6RP (S240/244) [dilution 1:2000], p-GSK-3β (S9) [dilution 1:1000], and total PRAS40 [dilution 1:1000], S6RP [dilution 1:2500], GSK-3β [dilution 1:1000] were acquired from Cell Signaling Technology. V5 probe (E10) [dilution 1:2000] and β-actin antibodies (C4) [dilution 1:5000] were acquired from Santa Cruz Biotechnology.

**Proliferation studies**. Ba/F3 cells were washed with PBS, resuspended in assay medium, counted, and 30,000 cells/well were seeded in triplicate for each of stable cell lines expressing WT or mutant AKT. Before counting, cell suspension was mixed for homogeneity, and 1 ml aliquots were counted in ViCell-XR every day of the 7-day study period.

**Immunofluorescence (IF) and antibodies**. MCF10a stable cells seeded in chamber slides were either untreated or treated with either DMSO or capivasertib or ARQ-092 (3 μM). Four hours post treatment, cells were fixed with 4% paraformaldehyde, permeabilized, blocked with 5% normal goat serum diluted in PBS, and incubated overnight with primary antibodies. The following day, cells were washed, incubated with secondary antibodies conjugated to fluorescent probes (Alexa Fluor 488 [dilution 1:400] and 594 [dilution: two drops per ml, as per instructions], ThermoFisher), and the slides were mounted using mounting media with DAPI (VectaShield).

**Drug treatment and cell viability assay**. Capivasertib was provided by Astra-Zeneca. MK-2206 and ARQ-092 were purchased from Selleck Chemicals. Compounds were dissolved in DMSO to yield a 10 mM stock, and diluted in assay medium to achieve the desired concentrations. MCF10a cells from stable lines expressing WT or mutant AKT were seeded in 96-well plates and treated with a range of drug concentrations in assay media, and cell viability was assessed 72 h post treatment using the Cell Titer Glo assay (Promega). Viability data were analyzed and $IC_{50}$ values derived using GraphPad Prism. Wortmannin was purchased from Selleck Chemicals and BYL-719 was kindly provided by the N. Rosen laboratory. MCF10a cells stably expressing AKT1 or AKT2 mutations were seeded in 6-well plates, treated with different concentrations of wortmannin or BYL-719 in complete growth media, and cell lysates were harvested 2 h post treatment.

**Thermal stability assays**. Approximately $2 \times 10^7$ cells were collected in 1 ml PBS containing Roche Complete Protease Inhibitor tablet. Cells were lysed by subjecting them to three freeze-thaw cycles. 100 μl of whole-cell lysates were aliquoted into PCR tubes and exposed to a thermal gradient from 40 to 64 °C for 3 min, and maintained at 25 °C for additional 3 min. The tubes were centrifuged at $20,000 \times g$ for 20 min at 4 °C. Clear lysates were then transferred to fresh eppendorf tubes, mixed with SDS loading buffer, and the relative stability of proteins of interest was analyzed by western blot.

**AKT1 allele structure modeling in open and closed conformations**. Protein structures for WT AKT1 in open and closed conformations were obtained by homology-based molecular modeling using experimentally-derived AKT1 structures as templates. An experimental structure for allosteric inhibitor-bound AKT1 (PDB ID: 3O96[22]) was used as the template for AKT1 in a closed-conformation. Two structures were used for AKT1 in an open conformation: AMP-PNP bound AKT1 (PDB ID: 4EKK[23]), and AKT1 in complex with an ATP-peptide conjugate bisubstrate (PDB ID: 6NPZ[39]). For each template, we modeled the complete WT AKT1 protein using the I-TASSER Suite for homology and ab initio folding-based protein structure prediction (version 5.1[40,41]), with the "restraint3" argument to specify the given model template. Of five models generated for WT AKT1 from each template, we used the models with the highest C-scores: 3O96: 0.23, 4EKK: −0.85, 6NPZ: −0.89, corresponding to TM-scores of 0.74, 0.61, 0.60, respectively, all of which met the minimum thresholds to ensure the model had correct topologies[42]. Models for AKT1 with E17K and E341K mutations were generated using UCSF Chimera (version 1.12)[43] to insert the mutation in the WT structure for each template. The AKT1 P68-C77 duplication structures were generated by first modeling the structure of AKT1 for each template with an insertion of 10 alanine amino acids between residues 77-78 using I-TASSER. Here, the "-restraint2" option was used to specify that the insertion aligned to the appropriate position between residues 77-78 in the template structure, resulting in modeled structures with C-scores=0.03, −1.27, −0.44, and TM-scores=0.72, 0.56, 0.66 for templates 3O96, 4EKK, 6NPZ, respectively. The 10-alanine sequence was then replaced with the correct amino acid sequence for the duplicated AKT1 P68-C77 residues using Modeller (v1.19)[44] called from within UCSF Chimera. To evaluate the quality of the AKT1 68-77 duplication models, we assessed the structural similarity of the modeled PH domain between the WT and duplication models using TM-scores from the TM-align software[45], which we then converted to multiple-hypothesis corrected P values[42]. This analysis revealed that the modeled PH domains in WT and duplication structures were significantly more similar than expected by chance for 3O96 and 4EKK-derived models (Q values: $6.3 \times 10^{-9}$, $6.8 \times 10^{-6}$, respectively). However, in the duplication model based on template 6NPZ, the PH domain was unstructured and inconsistent with that of the WT 6NPZ structure ($Q = 0.13$). We therefore retained models generated from 3O96 and 4EKK but excluded all 6NPZ-derived models from further analysis.

**Molecular dynamics simulations**. MD simulations were performed on structural models for WT AKT1, AKT1 E17K, AKT1 E341K, and AKT1 P68-C77dup, each generated by homology modeling using both 3O96 and 4EKK crystal structures as templates. All simulations were performed using GROMACS[41–43] (version 5.1.4)[46–48] with the Amber ff03 force field[49]. Each protein was placed in the center of a cubic box at least 1 nm from the box edge and solvated with SPC/E 3-point water molecules[50] and sodium ions to neutralize the negative charges of the proteins. Using structures derived from the 3O96 template, the resulting solutions were 79,985 atoms for WT AKT1 (7761 protein atoms, 11 Na atoms, and 24,071 water molecules); 79,990 atoms for the E17K mutant (7768 protein atoms, 9 Na atoms, and 24,071 water molecules); 79,990 atoms for the E341K mutant (7768 protein atoms, 9 Na atoms, and 24,071 water molecules); and 87,620 atoms for the P68-C77dup protein (7934 protein atoms, 9 Na atoms, and 26,559 water molecules). For the 4EKK template, the solutions were 80,252 atoms for WT AKT1 (7761 protein atoms, 11 Na atoms, and 24,160 water molecules); 80,266 atoms for AKT1 E17K (7,768 protein atoms, 9 Na atoms, 24,163 water molecules); 80,257 atoms for AKT1 E341K (7,768 protein atoms, 9 Na atoms, 24,160 water molecules); and 112,376 atoms for AKT1 P68-C77dup (7,934 protein atoms, 9 Na atoms, 34,811 water molecules). Periodic boundary conditions were used for all MD simulations. Prior to MD, the geometry of each system was relaxed through energy minimization with the steepest descent algorithm. The systems were then equilibrated to 300 K and 1 atm for 50 ps and their MD were simulated over 100 ns in steps of 2 fs. For all equilibration and MD processes, system coordinates were recorded every 10 ps. All bonds lengths were constrained using the LINCS algorithm[51,52], enabling 2 fs time steps during simulation. The Verlet scheme[53] was used to determine short-range nonbonded interactions using electrostatic and van der Waals interaction cutoffs of 1 nm each, and interacting neighbors were updated every 10 fs. Long-range electrostatic interactions were determined with the smooth particle mesh Ewald method[54]. The Velocity rescaling thermostat algorithm[55] was used to maintain the system at 300 K during MD simulations, with the protein and solution in separate temperature coupling groups as described previously. Pressure was maintained at 1 atm during MD simulations using the Parrinello-Rahman barostat algorithm[56].

**Structural impacts of AKT1 E17K and P68-C77dup variants**. Analysis and visualization of mutant AKT1 were performed using UCSF Chimera unless otherwise noted. Hydrogen bonds were determined with Chimera's FindHBond function using default parameters. Hydrophobic interactions were identified using the Find Clashes/Contacts function to search for carbon atoms among hydrophobic residues closer than four Å[57]. Time-averaged structures were generated from the MD simulation trajectories for each variant of AKT1 Chimera using the Average Structure utility, which were then used to generate average-residue-residue-distance maps. Protein residue relative solvent accessibilities were predicted for the time-averaged MD simulation structures using SSPIDER (version 2)[58].

To identify residues involved in the interaction between the PH and Kinase domains, we identified pairs of residues from the two domains (PH: residues 5–108, Kinase: 150–408) predicted to have hydrophobic interactions in the protein structure (using the closed, autoinhibited conformation of WT AKT1 based on PDB 3O96), based on the existence of their hydrophobic side chains, and at least six pairs of carbon atoms from each residue at a maximum distance apart of 4 Å (Supplementary Fig. 3). Among the pairs of interaction residues, many of the activating mutations arise either at one of these residues (L52, W80, L321) or proximal to them (L79, E322, V320). We then evaluated the effects of activating mutations on 37 pairs of strongly interacting residues in both the closed and open conformations, simulating AKT1 WT, E17K, and P68-C77dup alleles for 100 ns using as templates both PDB structure 3O96 (an allosteric inhibitor-bound AKT1, representing its closed conformation) and 4EKK (AMP-PNP bound, representing an open, ATP-bound conformation). We determined the root-mean-squared distances between the time-averaged positions for each pair of these interacting residues (Supplementary Fig. 3b). Analyses and heatmap visualization of the difference in inter-residue distance between AKT1 proteins were performed in R (version 3.4.1) and the ggplot2 library.

**Trial patient population**. This study was conducted at Memorial Sloan Kettering Cancer Center (MSKCC) between October 2017 and April 2020; the dates of enrollment for the first and last patients being 10/27/2017 and 7/24/2019, respectively. It was approved by the MSKCC Institutional Review Board (IRB) and performed in accordance with the Declaration of Helsinki. Eligible patients were 18 years of age or older with a pathologically confirmed diagnosis of a recurrent or metastatic advanced solid tumor with a somatic mutation in AKT1, AKT2, or AKT3 detected by prospective clinical sequencing. Estrogen receptor-positive (ER+) breast cancer patients must have progressed on fulvestrant and prostate cancer patients must have progressed on enzalutamide. Subjects were required to have measurable disease by either RECIST v1.1[59] (Eisenhauer), Prostate Cancer Clinical Trials Working Group 3 (PCWG3), or RANO criteria, and have an Eastern Cooperative Oncology Group (ECOG) performance score of 2 or less, life expectancy of 12 weeks or greater, and adequate organ and bone marrow function. Patients were excluded if they had ER+ breast cancer harboring an AKT1 E17K mutation (patient population tested in NCT01226316), diabetes mellitus type 1, diabetes mellitus type 2 requiring insulin or >2 oral hypoglycemic medications, glycosylated hemoglobin ≥8%, symptomatic brain metastases or spinal cord compression, recent anti-cancer therapy (see study protocol), or QTc >480 ms.

**Trial study design and treatment**. This investigator-initiated trial was an open-label, single institution, nonrandomized, pilot study that included three cohorts with up to 12 patients each: prostate cancer, ER+ breast cancer, and other solid tumors. No power calculation was performed as this was an exploratory, signal-seeking study of an uncommon class of somatic mutations. Informed consent was obtained from all participating patients as per guidelines, and as mentioned in the Protocol. Capivasertib was administered orally at the single agent recommended phase II dose on an intermittent schedule of 480 mg twice daily for four days on, three days off for "other" solid tumors, and 400 mg twice daily on this same schedule in combination with fulvestrant 500 mg IM per the approved schedule or enzalutamide 160 mg daily for breast cancer and prostate cancer patients, respectively. Dosing was on a 28-day treatment cycle and patients were evaluated at the start of every cycle for the first 24 weeks and every two cycles thereafter. Tumor imaging was obtained every two cycles. Patients remained on study treatment until progression of disease, intolerable adverse events, or withdrawal for any reason.

The primary endpoint was ORR- defined as the proportion of patients with a confirmed complete response (CR) or PR using either modified RECIST v1.1 or ≥

50% PSA decrease from baseline (in prostate cancer patients without visceral and/or nodal disease at baseline). Secondary endpoints included assessment of toxicity according to NCI common toxicity criteria (CTC) version 4.0, progression-free survival (PFS), and clinical benefit rate (CBR) defined as CR, partial response, or stable disease at 24 weeks.

**Trial statistical analysis**. The study aimed to accrue 12 patients in each cohort. The drug would be deemed to have preliminary evidence of clinically meaningful antitumor activity if three or more of 12 patients in a given cohort had a PR or CR. Unfortunately, the study was closed early due to the infrequency of enrolling mutations that drove slow accrual that did not meet enrollment milestones after a total of 12 patients were enrolled. Study closure was approved by the MSKCC IRB. Efficacy and safety data were therefore pooled and reported using descriptive statistics. For categorical variables, frequencies and percentages were utilized. Continuous variables were described using the mean, standard deviation, median, and range. PFS was calculated using the Kaplan-Meier method.

**Reporting summary**. Further information on research design is available in the Nature Research Reporting Summary linked to this article.

## Data availability

Source data are provided with this paper and are available here: https://doi.org/10.5281/zenodo.5111040[60]. All mutational data from the prospective sequencing available for download at http://cbioportal.org/. All other genomic and clinical data are available from public sources (MC3; https://gdc.cancer.gov/about-data/publications/mc3-2017) or accompanies the manuscript and is available as Supplementary information, including the clinical trial protocol. Other materials, code, and deidentified patient-level clinical data not otherwise presented in the Supplementary information is available upon request.

## Code availability

Source code for analyses is available at https://github.com/taylor-lab/akt1. All other code not otherwise presented in the Supplementary information is available upon request.

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

## Acknowledgements

We thank the members of the Taylor and Solit laboratories for discussion and support. This work was supported by National institutes of Health awards P30 CA008748, U54 OD020355 (B.S.T., D.B.S.), R01 CA207244 (D.M.H., B.S.T.), R01 CA204749 (B.S.T.), R01 CA229624 (D.B.S.); the American Cancer Society, Anna Fuller Fund, and the Josie Robertson Foundation (B.S.T.). Capivasertib was provided by AstraZeneca.

## Author contributions

T.S.B. and B.S.T. conceived the study. T.S.B., T.S., D.C., S.P., and S.P.G. designed and performed the experiments. A.N.G., M.T.C., E.I.G, M.T.A.D., and J.G. performed genomic and structural analyses. D.C. and S.P. assisted with mutant allele curation. T.S., M.H.R., W.A., D.M.H., A.M.S., D.B.S., and L.M.S. provided clinical specimens and clinical data. T.S.B. and B.S.T. wrote the manuscript with input from all authors.

## Competing interests

The authors declare the following competing interests: L.M.S. reports receiving research funding from AstraZeneca, Puma Biotechnology and Roche Genentech; honoraria from AstraZeneca and Pfizer; travel, accommodations, expenses from Puma Biotechnology, Roche Genentech and Pfizer; and consulting or advisory board activities for Roche Genentech and AstraZeneca. D.B.S. reports advisory board activities for Loxo Oncology, Pfizer, Illumina, Lilly Oncology, Vivideon, and Intezyne. D.M.H. reports receiving research funding from AstraZeneca, Puma Biotechnology, Loxo Oncology and personal fees from Atara Biotherapeutics, Chugai Pharma, Boehringer Ingelheim, AstraZeneca, Pfizer, Bayer, Debiophram Group, and Genentech. B.S.T. reports advisory board activities for Boehringer Ingelheim, Loxo Oncology (a wholly owned subsidiary of Eli Lilly), and honoraria and research funding from Genentech. All stated activities were outside of the work described herein. No other disclosures were noted.
