## [Peer Review File · Nature Communications]

Reviewers' comments:

Reviewer #1 (Remarks to the Author):

In this new paper, the authors analyzed somatic mutations of Akt1-3 isoforms in a large population of human cancers including primary and treatment naïve tumors, and then characterized Akt mutational hotspots including point mutations, in-frame insertion/deletion to understand their oncogenic effects and therapeutic relevance both in cells and clinical trials. The information of this paper is very useful for cancer signaling field and provide deeper understanding how the oncogenic Akt mutants can govern its pharmacological sensitivities. Even though the interest on the data the authors glossed over and skipped important details that should be included and presented. Below we will discuss some experimental points and also rise some questions and request for experiments to the authors:

1. Regarding to statistical analysis of mutations, they found some point mutations including Akt E17K mutant was the most common, which is not surprising to us from previous studies and from cBioPortal for cancer genomics (<https://www.cbioportal.org/>). However, they also discovered some more complex in-frame insertion and duplication mutants (rarely occurred but severely affect), perhaps this is the most interesting point in this paper. In addition, the authors further characterized the contribution of each of mutations on the phosphorylation of Akt and its downstream substrates in cell-based assays.

Surprisingly, they found Akt1 F55Y mutant did not elevate Akt phosphorylation but increased the phosphorylation of Akt substrates S6 and PRAS40. But the authors did not go further to explain this observation. It is necessary to perform in vitro kinase assays for at least mutations E17K, P68-C77 dup, F55Y to clarify the observations from cell works.

2. With a new discovery of in-frame insertion and duplication mutants (indels) that hyperactivate Akt signaling in cells, the authors used molecular dynamic simulation (MD) to predict any Akt conformational changes and suggested a more extended PH-out conformation of Akt induced by this type of mutation when compared to close conformation of WT Akt or partially open conformation of Akt E17K mutant. The consequence of the indels-induced Akt conformational change, clarified by the authors, is that Akt predominantly localized to the plasma membrane for its hyperactivation. However, it is unclear whether PIP3, the phospholipid that binds to Akt PH domain, is an effector in this hyperactivation process. An analysis of what happens if PIP3 synthesis is blocked and how much it accounts for binding affinity of PIP3 to Akt indels mutant versus E17K or WT Akt should be performed and added.

3. The authors analyzed the sensitivities of Akt indels mutants to allosteric and ATP-competitive inhibitors, but did not show IC50 as quantitative information for the readers. This information needs to be included and the methods described, particularly when the authors have discussed about the binding affinity of allosteric versus ATP-competitive inhibitors to Akt indels mutant.

4. Surprisingly, the author followed up their computational-cell (in silico and in vitro) studies with clinical trials, lacking translational step with animal studies for detailed pharmacological features on this new type of Akt mutation with ATP-competitive inhibitor AZD5363. Or if this data has been published elsewhere, the authors need to mention and discuss here.

5. The analysis using only 3O96 (PH-inhibitor part of kinase) instead of 3O96 and the other structures 4EKK and 6NPZ has led the authors into wrong conclusions. For example, the authors suggest that the computational analysis predicted the mutations to be unremarkable. These should be reanalyzed since for example E341K, since E341 is in salt bridge contact with Arg2 of the bisubstrate (6NPZ), a reverse

charge mutation such as EtoK will have a big impact either reducing the binding of the bisubstrate or GSK3 peptide or and or a local rearrangement of 341 are. THE citations to the PDB ID (4EKK and 6NPZ) and the corresponding papers should be included.

7. The authors used only 3O96 which although it has the pH domain it has an inhibitor bound which affects conformations of areas of the structure relevant to the data presented. An extended analysis including all structural data available should be included.

6.The citation to the structure of the AKT1 PH should also be cited 1H10. Moreover a more detailed elaboration of lines 90-94 on in what is based the proximity of phd-kd interface is required; as well as the difference in structural elements with 3O96 and 6NPZ.

7-Addition of figures analyzing pH domain residues R15-18 and KD in the 320ish in 2 of the conformations is required for full disclosure

Typos and minor corrections:

In methods a symbol before restraint line 326, 333 Etc

A space is needed between the numerals and the units example line 357, 350 etc

Line 365 change A for the correct symbol Å

Grammar on the sentence line 77-78 is not correct. We also expressed four AKTQ mutations that, while emerging rarely in patients, our computational analysis predicted were unremarkable.

May be should be “predicted as unremarkable”

The definition of MDS is not necessary, keep it as MD simulations and is less confusion to the reader

Reviewer #2 (Remarks to the Author):

The authors attempt to characterize the consequences of the long tail of AKT mutations that remain relatively uncharacterized. They have characterized the frequency of the aberrations in a very large data and also identified rare AKT indel mutations. This extend previous data from other studies including large studies from other groups based on TCGA data. The authors characterize a number of the mutations for signaling changes, cell growth inhibition and growth factor independent.

Concerns:

The authors use increase in AKT phosphorylation and downstream activation of the AKT pathway as the main readout for activity of mutants. This is reasonable but ignores evidence in multiple systems of neomorphic activity of other classes mutants and further of the potential for activation of other molecules by mutants. Further, there are discrepancies between AKT activation and transforming activity of several of these mutations (as referenced by the authors) and other studies demonstrating transforming activity of AKT mutations. The authors should extend their assays for all mutations to transformation of MCF10A and BAF3 as done for a subset of the mutations. Further, the key assay of in vivo growth should be performed as this captures information that is not available in in vitro phosphorylation assays. This would allow an analysis of the sensitivity of several of the mutations to the allosteric and catalytic inhibitors in this study. This would be particularly important for the specific L78_Q79Ins mutation in the illustrative patient.

The indel mutations and the W80 mutation being insensitive to the allosteric ARC and MK2206 inhibitor is very interesting. Why the W80 mutation is insensitive to the ARC compound and sensitive to MK2206 should be explored. This could at a minimum be studied by high resolution localization analysis. The authors should provide experimental validation of a mechanism to explain the insensitivity of the indels to the allosteric inhibitors. The authors claim there is an open structure from modeling but do not assess this experimentally. Why are they not responsive. The authors propose that the open configuration may induce membrane association that is insensitive to the allosteric inhibitors. The images of localization of AKT in figure 3f are not of sufficient resolution to determine whether the mutants and in particular the indels are fully associating with the cell membrane. Indeed, they do not look like the myr mutant controls. While there may be some cytosolic membrane association, the one indel presented is not solely associated with the cytosolic membrane. Further, the allosteric inhibitors and in particular MK2206 have been shown to inhibit AKT by moving AKT out of the membrane. Do they translocate the indels out of membrane association as compared to the point mutations. Additional indels and in particular the L78-Q79 ins should be assessed further.

Patient model: The authors attempt to justify the characterization of different tail mutations and their differential responses to allosteric and catalytic inhibitors by a study of a patient with a L78_Q79ins in AKT. There are several concerns with this study: First AKT inhibitors have reported activity in patients without AKT mutations (see studies from both Genentech and AstraZeneca including studies with AZD5363). Thus attributing the response of the patient with the L78_Q79ins mutation is at best a weak association. Secondly, the authors have not explored this mutation in all of the assays to demonstrate whether it is activating, transforming and selectively responsive to AZD5363. There is a L78-Q79Ins noted in multiple places in the manuscript but the insert is not indicated. These studies are required for use of this patient as an example. Without this data, the comments from the authors of "Overall, these results are consistent with the hyperactivation of AKT signaling induced by indel mutations and their hypersensitivity to AKT inhibition we observed in vitro, and suggest that activating non-E17K mutations in AKT1-3, including in-frame indels, may broaden the sensitizing biomarker of AKT inhibition, albeit in a mechanism-of-action-specific manner." is a major overinterpretation of the data. As an example, the L78-Q79Ins is not assessed in terms of selective sensitivity to the different AKT inhibitors in figure 4 a or b.

Minor comments

The manuscript requires editing. The first sentence in the abstract is concerning: Aberrant PI3K/Akt/mTOR signaling drives many human cancers, mediated in part by mutations in the AKT superfamily of serine/threonine kinases^{1,2}, a key effector of constitutive PI3K.

The term constitutive is not correct. The authors probably mean activated.

There are also concerns about the authors capturing previous references. For example " Candidate driver mutations in AKT1-3 were most common in breast cancers (4.3% of cases)." This ignores studies from Carpten from TGEN in 2007 and multiple follow on studies such as those from MDACC and the TCGA. Indeed, the frequencies in figure 1 reflect those already published in the TCGA. Further E17, L52, Q79 and other AKT mutations have previously been screened for activity by Kim et al Cancer Discovery

and Ng et al Cancer Cell. The consistency or not with the observations in these precedent manuscripts should be indicated.

Reviewer #3 (Remarks to the Author):

I read with great interest the manuscript by Bhattarai and colleagues titled, AKT mutant allele-specific activation dictates pharmacologic sensitivities. Overall, I found it compelling and novel, and I think it contributes to the field in a meaningful way. I have the following questions and concerns.

1. In figure 1 the authors present data on AKT mutations from 41,000 patients. There is some hand waving and references to prior reports that is really pretty thin. Please describe clearly the patient population of the current trial, including a standard table 1. Table one should include those important characteristics of the patients you are reporting on: different database sources and sizes, distributions of gender, age, tumor type, etc.

It is possible that this paper really has little to do with your cohort other than selecting a few of the many mutations. If that is true, then perhaps de-emphasize figure 1 and do not worry about the table 1 I describe above. However, if you are trying to describe the population impact of the mutations you are characterizing, then we need to have a clearer picture of your cohort.

2. Likewise for the mutations that you are considering - I recognize that it is non-trivial to combine mutations from multiple cohorts, but if you are going to present the distributions of AKT mutations in this manuscript as novel data, then we need more details on both the numerator and denominators you are considering. Both are unclear to me. How many mutations were called and exactly how did you arrive at that set? I cannot tell if there was simply a quality filtering of variants, or if there was some additional consideration of driver status based on your computational approach. For example, the lollipop figure 1d shows a lot of grey mutations, including some in grey that are clustered or redundant suggesting potential driver status simply by being recurrent. Where these included in your data for figure 1a? When you say "Candidate driver mutations in AKT1-3 were most common in breast cancers (4.3% of cases)" – does this only refer to your computationally preferred driver mutations, or all mutations that passed sequencing quality parameters.

3. If you ultimately only characterize a small set of variants, I think we should see the demographics on the tumors with those mutations and a repeat of figure 1a that just considered tumors with driver mutations.

4. I think it is quite reasonable to limit the number of mutations as the authors did. I am not clear the authors tell us specifically what fraction of all mutations are covered by the set of mutations that were generated experimentally. This relates to point #3 above. What disease generated these mutations, and does this therefore exclude some of the tumor types from being driven by AKT driver mutations?

5. I am likewise somewhat unclear as to how to interpret the remaining tail of mutations. Are we to officially brand these all passengers? The authors themselves founds at least a small number of unexpected results having made the set of mutations. Is there a recommendation for secondary screening, such as IHC for downstream targets or IHC-appropriate p-AKT antibody? This may be outside of the scope, but I feel this work might at least comment on the path forward for the remaining non-

canonical AKT mutants.

6. I recognize that this manuscript focuses on mutations, but at least in the introduction could we be reminded whether other mechanisms targeting AKT such as copy number amplification or other abnormality such as gene overexpression plays a role in AKT pathology. Do mutations occur in amplified gene segments such as EGFR or PIK3CA is this more commonly in copy neutral segments such as KRAS.

7. Again, while outside the scope of this manuscript, the background might provide some context on epistatic genetic alterations. Specifically, do we know if true AKT driver mutations are found in correlation or anticorrelation with other PIK3CA pathway mutations? Any other known concomitant or co-occurring mutations? Given this cohort, I would think that would be of some importance to assessing the likely efficacy of an AKT inhibitor and further drug development. Specifically, a drug-sensitive AKT mutant patient might still be sabotaged by downstream mutations if they were commonly found together.

8. For the indel mutations we would often see the genome alignments for novel mutations presented in the supplement such as an IGV figure. I suppose times have moved on, but when reporting a novel structural oncogenic variant, we would like to see the alignments and the nature of the sequence. This could be important for detection assays going forward.

9. There is one part of this study that I find truly unorthodox, which is the presentation of a single patient from a clinical trial. Granted this is an exceptional responder, but I think this is really not an appropriate way to present this story. I would exclude that case and report the clinical trial separately with all of the routine clinical trials reporting structures. Trial design. Patient assignment. Doses delivered. Toxicity. Etc. Perhaps that could be co-published with this science? I checked clinicaltrials.gov and this study is still listed as open. I do not think it is generally ethical to report results on a trial that is still ongoing. I would ask the editors for their guidance on this.

** See Nature Research's author and referees' website at www.nature.com/authors for information about policies, services and author benefits

Point by point response to Reviewers

Reviewer #1 commented that “*The information of this paper is very useful for cancer signaling field and provides a deeper understanding of how the oncogenic Akt mutants can govern its pharmacological sensitivities.*” Nevertheless, they also identified “*some experimental points and also raise some questions and request for experiments to the authors*”, which we address below:

Regarding statistical analysis of mutations, they found some point mutations including Akt E17K mutants were the most common, which is not surprising to us from previous studies and from cBioPortal for cancer genomics (<https://www.cbioportal.org/>). However, they also discovered some more complex in-frame insertion and duplication mutants (rarely occurred but severely affected), perhaps this is the most interesting point in this paper. In addition, the authors further characterized the contribution of each mutation on the phosphorylation of Akt and its downstream substrates in cell-based assays. Surprisingly, they found Akt1 F55Y mutant did not elevate Akt phosphorylation but increased the phosphorylation of Akt substrates S6 and PRAS40. But the authors did not go further to explain this observation. It is necessary to perform in vitro kinase assays for at least mutations E17K, P68-C77dup, F55Y to clarify the observations from cell works.

Response: We thank the Reviewer for positive comments about the in-frame insertion and duplication mutants we characterized here. In response to the comment regarding *AKT1 F55Y*, we have now performed a kinase assay as suggested using protein extracts obtained from *AKT1 WT*, *E17K*, *F55Y*, *P68-C77dup*, and *Myr-AKT1 WT* cells. *AKT1 E17K* showed a modest increase in kinase activity over *WT*, as assessed by the levels of p-GSK3 α . By contrast, the *P68-C77dup* mutant had far greater kinase activity than *E17K* (as did *Myr-AKT1*; included as a positive control) (see Figure below). These results are consistent with our observations throughout the study. The kinase activity of *AKT1 F55Y* was again ambiguous, with both V5 and *AKT*-specific antibody pull-down results indicating modestly (if any) greater kinase activity compared to *WT*, perhaps similar to the *E17K* allele, albeit at far lower expression in cells (see Figure). The significantly lower amount of protein being immunoprecipitated in both cases for *F55Y* complicates establishing a definitive result. Overall, as highlighted by the Reviewer, the *F55Y* results throughout our study—including the elevated phosphorylated S6 and PRAS40 (**Fig. 2a**), increased sensitivity of mutant cells to both ATP-competitive and allosteric *AKT* inhibitors similar to the levels achieved by definitive activating alleles (**Fig. 4a-b** and **Supplementary Fig. 6a**), and the new kinase activity results shown below—all suggest this allele is likely a modestly activating missense mutation despite the apparent absence of increased p-*AKT* levels in our experiments with MCF10a cells. Moreover, in our extended studies using Ba/F3 stable cells (see below), *AKT1 F55Y* had superior expression and both p-*AKT* signals (both T308 and S473) were detectable by western blot (see Figure below in response to Reviewer #2). Consequently, the *F55Y* findings appear to result more as a function of expression level than of a non-canonical *AKT* signaling component. Nevertheless, we agree that a deeper mechanistic understanding of how *AKT1 F55Y* achieves this pathway activation via potential kinase-dependent or independent functions is a noteworthy line of inquiry but believe this is outside the scope of the present study as our investigation is focused on revealing the unique structural, functional, and therapeutic implications of complex indel mutants in *AKT*. In response, we have nevertheless updated the main text of the manuscript to mention these new data in Ba/F3 cells and the difference in signaling of the *AKT1 F55Y* mutant based on expression levels.

With a new discovery of in-frame insertion and duplication mutants (indels) that hyperactivate Akt signaling in cells, the authors used molecular dynamic simulation (MD) to predict any Akt conformational changes and suggested a more extended PH-out conformation of Akt induced by this type of mutation when compared to close conformation of WT Akt or partially open conformation of Akt E17K mutant. The consequence of the indels-induced Akt conformational change, clarified by the authors, is that Akt predominantly localized to the plasma membrane for its hyperactivation. However, it is unclear whether PIP3, the phospholipid that binds to Akt PH domain, is an effector in this hyperactivation process. An analysis of what happens if PIP3 synthesis is blocked and how much it accounts for binding affinity of PIP3 to Akt indels mutant versus E17K or WT Akt should be performed and added.

Response: To address this question, we used MCF10a cells stably expressing WT AKT1/2, E17K or one of multiple indels and treated these with either Wortmannin (a non-specific covalent inhibitor of the PI3K's) or BYL-719 (a PIK3CA inhibitor). We sought to evaluate their ability to activate AKT when PIP3 formation is inhibited. As shown in the Figure below, increasing concentrations of both agents led to diminished p-Akt in both WT and E17K cells, suggesting their dependence on PIP3 formation for pathway activation. By contrast, activation by the indel mutants was independent of PIP3 conversion, as indicated by their unaffected p-Akt levels. AKT1/2 indel-mutant cells can therefore hyperactivate the PI3K pathway independent of PIP3 binding. We are grateful for this query from the Reviewer as this finding strengthens our results significantly. Accordingly, we have added these data as a new **Supplementary Fig. 5** and revised the main text.

The authors analyzed the sensitivities of Akt indels mutants to allosteric and ATP-competitive inhibitors, but did not show IC₅₀ as quantitative information for the readers. This information needs to be included and the methods described, particularly when the authors have discussed the binding affinity of allosteric versus ATP-competitive inhibitors to Akt indels mutant.

Response: We have now included the data from the drug viability studies involving WT AKT1/2 and the different mutants, including all IC₅₀ values (see below), as a new **Supplementary Table 2** and revised the manuscript accordingly. The cell viability assays were described in depth in the Methods section and all IC₅₀s and curve fitting were calculated using GraphPad Prism.

Gene	Mutant	IC ₅₀ (uM)			Status[a]
		AZD5363	ARQ092	MK2206	
AKT1	WT	40.1	5.8	9.5	Wildtype
AKT1	R15Q	125.6	6.1	6.5	NA
AKT1	E17K	7.2	3.5	5.7	AM
AKT1	W22R	61.2	11.8	19.2	NA
AKT1	E40K	9.5	3.1	2.5	AM
AKT1	D44N	50.8	9.0	6.7	NA
AKT1	R48H	55.4	6.5	7.5	NA
AKT1	L52R	15.5	6.3	6.8	AM
AKT1	F55Y	11.5	5.0	7.7	AM
AKT1	T65-I75dup	4.8	13.8	52.5	AI
AKT1	E66-Q79dup	4.1	33.0	78.3	AI
AKT1	P68-C77dup	2.7	41.9	88.3	AI
AKT1	Q79K	5.8	4.7	3.3	AM
AKT1	W80R	8.8	6.6	61.8	AM
AKT1	E267G	84.7	8.5	10.9	NA
AKT1	D323G	9.1	4.0	3.9	AM
AKT1	E341K	32.9	9.3	9.9	NA
AKT1	R370C	61.3	6.4	5.0	NA
AKT1	E464K	60.5	7.3	4.9	NA
AKT2	WT	23.9	3.7	-	Wildtype
AKT2	E17K	8.1	4.4	-	AM
AKT2	60_75dup	6.4	4.1	-	AI
AKT2	67_78dup	6.6	5.3	-	AI
AKT2	68_80dup	5.3	24.2	-	AI
AKT2	76_84dup	4.9	22.3	-	AI
AKT2	78_79ins	4.4	16.9	-	AI

[a] AM, activating missense; AI, activating indel; NA, not activating

Surprisingly, the author followed up their computational-cell (*in silico* and *in vitro*) studies with clinical trials, lacking translational step with animal studies for detailed pharmacological features on this new type of Akt mutation with ATP-competitive inhibitor AZD5363. Or if this data has been published elsewhere, the authors need to mention and discuss here.

Response: The Reviewer raises an excellent point regarding the acceleration to clinical translation. First, we note that detailed pharmacological studies of the ATP-competitive inhibitor AZD5363 have already been performed in animals (Davies BR, et al. *Mol Cancer Ther* 2015). Second, our group has tested the safety and efficacy of AZD5363 in an early-phase clinical trial of patients with solid tumors harboring *ATK1* E17K, which came after the aforementioned detailed *in vivo* pharmacological studies (Hyman DM, et al. *J Clin Oncol* 2017). Third, our laboratory has published new approaches for accelerating clinical hypothesis testing, including an exploratory framework by which computational weight of evidence alone was utilized in real-time to prioritize treatment-refractory patients harboring novel hotspot mutations of uncertain clinical significance for studies of molecularly targeted therapies (Hyman DM, et al. *Cell* 2017). We have tested this approach in a subset of patients, whereby clinical response rather than laboratory interrogation was employed as the most expedient approach for clinical validation of mutant alleles of unknown function arising in genes for which targeted therapies are already available (Chang MT, et al. *Cancer Discov* 2018; Hyman DM et al. *Nature* 2018). Our current study builds on our extensive understanding and clinical testing of AZD5363 to extend this already tested framework to the full and formal design of an investigator-initiative clinical trial, the full results of which are now reported here (see below). We hope the Reviewer agrees this

represents an acceleration of the clinical translation of our research findings in a manner that is conservative based on our extensive prior work testing AKT inhibition in AKT-mutant solid cancers.

The analysis using only 3O96 (PH-inhibitor part of kinase) instead of 3O96 and the other structures 4EKK and 6NPZ has led the authors into wrong conclusions. For example, the authors suggest that the computational analysis predicted the mutations to be unremarkable. These should be reanalyzed since for example E341K, since E341 is in salt bridge contact with Arg2 of the bisubstrate (6NPZ), a reverse charge mutation such as E to K will have a big impact either reducing the binding of the bisubstrate or GSK3 peptide or and or a local rearrangement of 341 are. THE citations to the PDB ID (4EKK and 6NPZ) and the corresponding papers should be included.

Response: We thank the Referee for this useful suggestion. To clarify, the original text referred to E341 as unremarkable based on computational studies that included human tumor recurrence and sequence paralogy analyses and not the detailed MD simulations that followed later in the narrative to explore the function of the indel mutants. We apologize for this lack of clarity and we have modified the revised text accordingly.

While we do not believe that we drew the wrong conclusions from our analysis, we nevertheless agree it would be valuable to see how our analysis differs between AKT1 in open and closed conformations. We had originally used 3O96 as the template for our structural analyses for two reasons. First, 3O96 is among the few established AKT1 structures to include both PH and kinase domains and is not bound to covalent inhibitors. Neither 4EKK nor 6NPZ possess the PH domain in its entirety (see below Figure panel A). As structural modeling can be inaccurate, we reasoned that using a structure where both PH and kinase domains are available would improve the accuracy of the AKT1 model and the predicted effects of mutations. Second, because 3O96 is allosteric inhibitor-bound, it is necessarily in a closed conformation. As we hypothesized that the duplication mutants promote an open-like conformation, modeling their effects in a closed conformation was necessary. Nevertheless, we did compare the effects of these mutants in AKT1 in open conformation. We modelled AKT1 WT and the E17K, E341K, and P68-C77dup alleles in all three structures: 3O96, 4EKK, and 6NPZ. Unfortunately, the P68-C77dup model generated using 6NPZ was unstable due to the aforementioned need to model the entire PH domain, which in this structure was too different to meaningfully compare to WT, see below Figure panels B). As the PH domain in 4EKK remained structured in both WT and P68-C77dup, we chose 4EKK as the representative structure of AKT1 in an ATP-bound conformation.

The resulting analysis (see Figure below) indicated that P68-C77dup had a more profound structural effect on AKT1 in closed conformation (3O96; allosteric inhibitor-bound) as compared to ATP-bound AKT1 in open conformation (4EKK). This is consistent with the predicted role of the duplication mutant in promoting an open-like conformation. As the Referee suggested, E341K does indeed result in a modest structural deformation between PH and kinase domain

interaction akin to that achieved by E17K when AKT1 is in closed conformation. Notably, neither E17K nor E341K had any effect on the PH-kinase domain interface in the 4EKK template, which suggests that these mutations are likely functional in only the closed conformation and is consistent with their effect of promoting active-conformation AKT signaling. In response, we have modified the manuscript to include an expanded results and Methods section and new **Supplementary Fig. 3a** describing the modeling using the 4EKK structure in open conformation, added the citations to other AKT structures specified by the Reviewer, and adjusted our language about E341 to more accurately reflect our original rationale for its inclusion.

The authors used only 3O96 which although it has the pH domain it has an inhibitor bound which affects conformations of areas of the structure relevant to the data presented. An extended analysis including all structural data available should be included.

Response: We thank the Referee for this suggestion, and agree that it is important to investigate the effects of oncogenic mutations on AKT1 in its different conformations. As we described above, we used 3O96 due to its structural completeness compared to other published models (contains both PH and kinase domains), and to ensure we evaluated our hypothesis on the effect of duplication mutants on AKT1 in closed (inactive) conformation achieved by 3O96 which is indeed allosteric inhibitor-bound. Nevertheless, per the suggestion of the Referee, we repeated our structural analysis using the 4EKK template, which is an ATP-analog bound form of AKT1 representing an open conformation, the results of which are described above and now in the revised manuscript and new **Supplementary Fig. 3a**. While we also attempted this analysis using the 6NPZ structure of AKT1 bound to a bisubstrate (another open conformation template), homology modeling failed to produce a reliable structure for the PH domain, which was not published by the original study authors as part of this structure. We have now updated the manuscript to describe the results of structural modeling in the open conformation 4EKK model. We are grateful to the Reviewer for the suggested analysis as the results strengthen our findings overall.

The citation to the structure of the AKT1 PH should also be cited 1H10. Moreover a more detailed elaboration of lines 90-94 on in what is based the proximity of phd-kd interface is required; as well as the difference in structural elements with 3O96 and 6NPZ.

Response: We thank the Reviewer for these suggestions. We have now added the appropriate citation for the 1H10 structure. Regarding our statement: “the majority of [activating non-E17K

missense mutations] lie in close proximity to the PHD-KD interface”, this refers to our analysis of the interface of the PH and Kinase domains of AKT1 in a closed, autoinhibited conformation using a model for complete wild-type AKT1 based on PDB 3O96. To identify residues involved in the interaction between the two domains, we identified pairs of residues from the PH and Kinase domains (PH: residues 5-108, Kinase: 150-408) predicted to have hydrophobic interactions in the protein structure, based on (1) the existence of their hydrophobic side chains, and (2) at least 6 pairs of carbon atoms from each residue at a maximum distance apart of 4Å. The results from this analysis are shown in the figure below. Among the pairs of interacting residues, many of the activating mutations arise either at one of these residues (L52, W80, L321) or proximal to them (L79, E322, V320). In the revised manuscript, we have amended the methods accordingly to clarify these important details.

Addition of figures analyzing pH domain residues R15-18 and KD in the 320ish in 2 of the conformations is required for full disclosure

Response: As suggested by the referee, there are important hydrophobic interactions between the PH and kinase domains when in a closed and auto-inhibitory conformation, such as between residues Y18 and L321, but also other residue pairs at the interface of the domains (e.g. W80-K268, I84-F309). We have now evaluated the effects of activating mutations on these interactions (37 pairs of strongly interacting residues in the PH and kinase domain in total) in both the closed and open conformations. We simulated each of AKT1 WT, E17K, E341K, and P68-C77dup alleles for 100ns using as templates both PDB structure 3O96 (an allosteric inhibitor-bound AKT1, representing its closed conformation) and 4EKK (AMP-PNP bound, representing an open, ATP-bound conformation), and then determined the root-mean-squared-distances (RMSD) between the time-averaged positions for each pair of these interacting residues. From these simulations, we observed that when initially in a closed conformation, the P68-C77dup mutation resulted in a significant increase in the distances between interacting residues (median RMSD=10.5Å) compared to WT (median RMSD=8.5Å) ($P = 0.0002$, two-sided unpaired Wilcoxon rank sum test, see Figure below). Notably, E341K and E17K did not result in a significant difference in RMSD compared to wild-type ($P = 0.3$ for each allele compared to WT). Conversely, when simulations began with AKT1 in an open conformation, the interacting residues in each allele had large initial distances (median RMSD > 14Å). In this conformation, the P68-C77dup had a weaker effect on the distance between interacting residues (median RMSD=19.6Å compared to 14.4Å in WT, $P=0.03$), and as before the E17K and E341K did not significantly affect the interacting residues. Collectively, these simulations indicate that one consequence of the P68-C77 duplication is to push apart residues normally involved in PH and Kinase-domain interactions promoting an open-like conformation, thus reducing the capacity for auto-inhibition. We appreciate the suggestion of the Reviewer and have now modified the

revised manuscript (both the main text, Methods section, and added a new **Supplementary Fig. 3b**) to include these expanded analyses and results.

In methods a symbol before restraint line 326, 333 Etc

Response: This was an issue introduced by manuscript conversion during submission and has now been corrected in the revised manuscript.

A space is needed between the numerals and the units example line 357, 350 etc

Response: This is now corrected in the revised manuscript.

Line 365 change A for the correct symbol Å

Response: This is now corrected in the revised manuscript.

Grammar on the sentence line 77-78 is not correct. We also expressed four AKTQ mutations that, while emerging rarely in patients, our computational analysis predicted were unremarkable.

May be should be "predicted as unremarkable"

Response: We have now modified this sentence as suggested in the revised manuscript.

The definition of MDS is not necessary, keep it as MD simulations and is less confusion to the reader

Response: We have now modified this definition as suggested by the Reviewer in the revised manuscript.

Reviewer #2 commented that our study "extend previous data from other studies including large studies from other groups based on TCGA data" and that we "characterize a number of the mutations for signaling changes, cell growth inhibition and growth factor independent." They nevertheless raised several points which we address below:

The authors use increase in AKT phosphorylation and downstream activation of the AKT pathway as the main readout for activity of mutants. This is reasonable but ignores evidence in multiple systems of neomorphic activity of other classes of mutants and further of the potential for activation of other molecules by mutants. Further, there are discrepancies between AKT activation and transforming activity of several of these mutations (as referenced by the authors) and other studies demonstrating transforming activity of AKT mutations. The authors should extend their assays for all mutations to transformation of MCF10A and BAF3 as done for a subset of the mutations. Further, the key assay of *in vivo* growth should be performed as this captures information that is not available in *in vitro* phosphorylation assays. This would allow an analysis of the sensitivity of several of the mutations to the allosteric and catalytic inhibitors in this study. This would be particularly important for the specific L78_Q79Ins mutation in the illustrative patient.

Response: Based on the reviewer's suggestions, we extended our study to a broader panel of AKT1 and AKT2 mutants by generating BaF3 stable lines expressing a wide variety of missense and indel mutants, like we had in MCF10a cells, and evaluated them both biochemically and phenotypically. We assessed these mutants for their ability to stimulate PI3K pathway activation in the absence of IL3 by western blot, as well as their ability to transform Ba/F3 cells to IL3-independent growth (see Figure below). AKT1/2 indel mutants had elevated levels of phosphorylated AKT as well as of downstream targets as assessed by western blot, indicating robust pathway activation. These mutants also promoted IL3-independent proliferation to much greater degree compared to the rather modest levels induced by other activating missense mutants. The degree of IL-3-independent transformation induced by indels compared to missense mutants was approximately proportional to the degree of pathway activation we observed. We therefore posit that the phenotypic effect exerted by the mutants we tested is a consequence of kinase activation rather than an atypical kinase-independent function of mutant AKT. We have added these significantly expanded Ba/F3 findings to the revised main text and adjusted **Figure 2** accordingly.

Regarding the suggestion of *in vivo* growth assays, the cell culture model that we used for our *in vitro* studies (MCF10a) is not appropriate for *in vivo* experiments because, as primary breast

epithelial cells, they are incapable of forming xenografts in mice. We therefore generated MCF-7 breast cancer cells in which the endogenous mutant PIK3CA E545K allele had been converted back to wildtype. We generated new MCF-7(WT PIK3CA) stable cell lines expressing ATK1 WT, E17K and P68-C77dup as well as AKT2 WT, E17K, and L78-Q79ins. We had scheduled our experiments and were ready to implant the cells in collaboration with the Antitumor Assessment Core Facility at our institution. However, this coincided with the statewide New York stay-at-home order and COVID-19 shutdown of all research and laboratory operations at our Center, preventing us from performing these studies in a timely manner. Institutional restrictions on key lab-based operations affected all aspects of research operations, non-essential research personnel activities, reagent supply chains including mice, and animal facility access, ultimately preventing the completion of in vivo studies despite our best efforts. We hope the Reviewer appreciates these extraordinary factors and we hope that in lieu of these studies, the significant expansion of key preclinical and clinical data from patients enrolled on the accompanying clinical trial serves as key validating data.

The indel mutations and the W80 mutation being insensitive to the allosteric ARC and MK2206 inhibitor is very interesting. Why the W80 mutation is insensitive to the ARC compound and sensitive to MK2206 should be explored. This could at a minimum be studied by high resolution localization analysis. The authors should provide experimental validation of a mechanism to explain the insensitivity of the indels to the allosteric inhibitors. The authors claim there is an open structure from modeling but do not assess this experimentally. Why are they not responsive?

Response: We appreciate the positive comments of the Referee and have now investigated these results further. Regarding the Referee's first point about W80 mutations, allosteric inhibitors of the AKT kinase bind to a pocket formed at the interface between PH and kinase domains, thereby locking the kinase domain in an inactive conformation that prevents ATP binding, membrane association, and activation loop phosphorylation. Critical for such inhibitor binding is both an intact PH domain and the W80 residue (Green CJ et. al. *J Biol Chem.* 2008; Calleja V etl al., *PLoS Biol.* 2009; Wu WI et. al. *PLoS ONE.* 2010). Hence, missense mutations in W80, or other cancer-associated mutations that disrupt the PH-kinase domain interactions, render allosteric AKT inhibitors less effective (Green CJ et. al. *J Biol Chem.* 2008; Parikh C et. al. *Proc Natl Acad Sci USA.* 2012). Although both MK2206 and ARQ092 are allosteric AKT inhibitors and inactivate the kinase by stabilizing its closed conformation, only the latter (ARQ092) further deactivates AKT by dephosphorylation of the membrane-localized active form. Moreover, ARQ092 shows better binding dynamics with both WT AKT and the E17K mutant, and has a higher potency than MK2206. Unlike MK2206, ARQ092 retains its ability to inhibit membrane translocation, phosphorylation, and kinase activation even in the presence of growth factors that push the PH domain out of closed conformation (Yu Y et. al. *PLoS One.* 2015). Collectively, these data indicate ARQ092 has increased potency and effectiveness despite mutational changes that alter PH-kinase domain interactions and compromise, to different extents, the structural integrity between these two domains. It is perhaps owing to a combination of these factors that ARQ092 is more effective against the AKT1 W80R mutant. While such a detailed and nuanced discussion of these differences between distinct allosteric inhibitors of AKT are outside the scope of the present manuscript, we have nevertheless updated the main text of the manuscript to elaborate on the difference among these inhibitors in their sensitivity in W80-mutant cells.

In response to the Referee's second point about experimental validation for the proposed mechanism of indel mutant insensitivity to allosteric inhibitors, we have now performed thermal stability assays for AKT1 WT, E17K, and multiple indel mutations observed in patients. Using

thermal shift assays (TSAs), we have now experimentally validated the open conformation induced by AKT1 indels as was suggested by our MD simulation analysis (see Figure below). TSA involves exposing natively folded proteins to increasing heat to determine the melting temperature (T_m) and protein stability, and is therefore indicative of molecular interactions that play a stabilizing role in structural organization. Proteins with open and unstable structures begin to denature and aggregate at lower temperatures (hence display lower melting temperature, T_m), whereas those with properly folded and more stable conformation denature only at higher temperatures (and thus have higher T_m) (Jafari et. al. *Nature Protocols*. 2014; Huynh & Partch. *Curr Protoc Protein Sci*. 2015). Both our group and others have used similar TSAs to evaluate the structural consequences of mutations in oncogenes (Vasan, N et al. *Science*. 2019; Croessmann S et. al. *Clin Cancer Res*. 2018). Here, we show that AKT1 harboring indel mutations readily denature at lower temperatures, likely due to their open conformation and structural destabilization resulting from the loss of stabilizing inter-domain interactions as suggested by our MD simulation studies. By contrast, closed conformation WT AKT1 remains stable through a higher temperature range. E17K appears to be less stable than the WT, but more so than the indels, consistent with modest structural changes suggested by our MD simulation analysis. Overall, these new results significantly strengthen our conclusions and we have modified the main text of the manuscript accordingly as well as added a new **Supplementary Fig. 9** showing these new results. We are grateful for the suggestion by the Referee.

The authors propose that the open configuration may induce membrane association that is insensitive to the allosteric inhibitors. The images of localization of AKT in figure 3f are not of sufficient resolution to determine whether the mutants and in particular the indels are fully associating with the cell membrane. Indeed, they do not look like the myr mutant controls. While there may be some cytosolic membrane association, the one indel presented is not solely associated with the cytosolic membrane. Further, the allosteric inhibitors and in particular MK2206 have been shown to inhibit AKT by moving AKT out of the membrane. Do they translocate the indels out of membrane association as compared to the point mutations. Additional indels and in particular the L78-Q79 ins should be assessed further.

Response: To address this question, we performed further immunofluorescence for both missense and indel mutants upon treatment with all three either ATP competitive or allosteric inhibitors. Treatment with allosteric inhibitors MK2206 and ARQ092 resulted in significant reduction in membrane association and phosphorylation for AKT1 WT and E17K (see Figure below). However, treatment with ATP competitive inhibitor AZD5363, causes paradoxical hyperphosphorylation of AKT, as previously described (Okuzumi T et. al. *Nat Chem Biol*. 2009 Jul; 5(7) 484-493). By contrast, membrane association and phosphorylation status of AKT1 P68-C77dup remained unchanged after treatment with the allosteric inhibitors (see below, far

right). These data, together with the absence of pathway inhibition we observed in **Fig. 4d**, further substantiates the inability of allosteric inhibitors to inhibit AKT indels. We thank the Reviewer for this suggestion and we have revised the main text of the manuscript accordingly and provided these data as new **Supplementary Fig. 8**.

Regarding the last point, while we did demonstrate membrane localization, pathway activation, and *in vitro* drug sensitivity for *AKT2* indels, their transforming potential was not evaluated in the original submission. In response, we have now performed these experiments and show that *AKT2* indels potently transform (independent of IL3) BaF3 cells (see prior response). We have now revised the manuscript to include these data as new **Fig. 2e**.

Patient model: The authors attempt to justify the characterization of different tail mutations and their differential responses to allosteric and catalytic inhibitors by a study of a patient with a L78_Q79ins in AKT. There are several concerns with this study: First AKT inhibitors have reported activity in patients without AKT mutations (see studies from both Genentech and AstraZeneca including studies with AZD5363). Thus attributing the response of the patient with the L78_Q79ins mutation is at best a weak association. Secondly, the authors have not explored this mutation in all of the assays to demonstrate whether it is activating, transforming and selectively responsive to AZD5363. There is a L78-Q79Ins noted in multiple places in the manuscript but the insert is not indicated. These studies are required for use of this patient as an example. Without this data, the comments from the authors of “Overall, these results are consistent with the hyperactivation of AKT signaling induced by indel mutations and their hypersensitivity to AKT inhibition we observed in vitro, and suggest that activating non-E17K mutations in AKT1-3, including in-frame indels, may broaden the sensitizing biomarker of AKT inhibition, albeit in a mechanism-of-action-specific manner.” is a major overinterpretation of the data. As an example, the L78-Q79Ins is not assessed in terms of selective sensitivity to the different AKT inhibitors in figure 4 a or b.

Response: For clarity, all references to the *AKT2* L78-Q79ins mutant throughout the manuscript refers to the L78_Q79ins(HANTFVIRCL) in-frame insertion mutation, and was shortened using standard nomenclature for brevity. Regarding the first point, we agree that the activity of AZD5363 has been reported in patients without AKT mutations (Lin J et. al. *Clin Cancer Res.* 2013 Apr 1;19(7):1760-72; Davies BR et. al. *Mol Cancer Ther.* 2012 Apr; 11(4) 873-887). However, these favorable responses were associated with activating mutations in *PIK3CA* and/or loss-of-function mutations in *PTEN*, both of which hyperactivate AKT and

aberrantly stimulate PI3K signaling. In addition, these studies discovered that *PIK3CA* kinase domain mutations, but not helical domain mutations, were significantly associated with increased sensitivity to AZD5363. Conversely, mutations in RAS and *BRAF* have been associated with increased resistance to both AZD5363 and GDC-0068 (another ATP competitive AKT inhibitor) (same citations as above). We believe these data, in concert with our own prior work establishing the clinical efficacy of this drug in *AKT1* E17K-mutant solid cancers (Hyman DM, et al. *J Clin Oncol*, 2017), reinforce the specificity of clinical activity in patients with PI3K pathway hyperactivation. Moreover, as mentioned in the main text, this metastatic prostate cancer patient harboring the *AKT2* L78_Q79ins(HANTFVIRCL) mutation was enrolled in the trial having previously responded to, but then progressed on, enzalutamide therapy. Given that our 1) tumor sequencing of a pre-treatment biopsy indicated no other driver mutations in effectors of PI3K pathway signaling were present other than the said *AKT2* indel, and 2) *in vitro* data showed that the *AKT2* indel is potently activating in both MCF10a and Ba/F3 cells (**Fig. 2c-d**), induces IL3-independent proliferation (**Fig. 2e**), and induces hypersensitivity to AZD5363 treatment (**Supplementary Fig. 7**), we believe these collectively suggest *AKT2* L78-Q79ins(HANTFVIRCL) was indeed the AZD5363-sensitizing event in this prostate cancer patient. Regarding the second point, we did confirm this allele was both activating and sensitized cells to AZD5363 (**Fig. 2c-d** and **Supplementary Fig. 7**, respectively). These *in vitro* drug sensitivity experiments were performed in our MCF10a cell culture model for both *AKT2* L78-Q79ins as well as other *AKT2* indels with both the ATP competitive and allosteric inhibitors. Similar to the *AKT1* indels we tested, these *AKT2* mutants hyper-sensitized cells to treatment with the ATP-competitive AKT inhibitor AZD5363. To ensure this is clearer and more evident to both the Reviewer and all readers, we revised the manuscript to provide more explicit pointers to these Supplementary Results in the main text and we also refer to the full mutant insertion.

The manuscript requires editing. The first sentence in the abstract is concerning: "Aberrant PI3K/Akt/mTOR signaling drives many human cancers, mediated in part by mutations in the AKT superfamily of serine/threonine kinases1,2, a key effector of constitutive PI3K." The term constitutive is not correct. The authors probably mean activated.

Response: We have now modified this sentence and others in the revised manuscript to ensure accuracy.

There are also concerns about the authors capturing previous references. For example "Candidate driver mutations in AKT1-3 were most common in breast cancers (4.3% of cases)." This ignores studies from Carpten from TGEN in 2007 and multiple follow on studies such as those from MDACC and the TCGA. Indeed, the frequencies in figure 1 reflect those already published in the TCGA. Further E17, L52, Q79 and other AKT mutations have previously been screened for activity by Kim et al Cancer Discovery and Ng et al Cancer Cell. The consistency or not with the observations in these precedent manuscripts should be indicated.

Response: We agree that prior work has sought to delineate the frequency of various AKT mutations using retrospective data in cancer, and we do cite work including Carpten, et al. 2007. Nevertheless, we believe there is value in refining the frequency of the most common AKT mutations (E17K) in a far larger and diverse (from the perspective of cancer types and disease stages profiled) sample size than has been studied previously, including from TCGA. Moreover, half of all data presented here is prospectively collected clinical sequencing data from active metastatic cancer patients in whom these data are used to guide treatment strategies. These data represent, therefore, a population of patients from whom clinical trial enrollments are drawn, including for those testing AKT inhibitors. This far larger sample size also better facilitates our goal of mining the tail of less common driver mutations in *AKT1-3*. However, to

ensure we are comprehensive in referencing past observations, we have now added greater context and the aforementioned citations along with comparison of our results with prior observations in the field to the revised manuscript.

Reviewer #3 commented that they read our study “with great interest” and that “Overall, I found it compelling and novel, and I think it contributes to the field in a meaningful way.” They had the following questions:

In figure 1 the authors present data on AKT mutations from 41,000 patients. There is some hand waving and references to prior reports that are really pretty thin. Please describe clearly the patient population of the current trial, including a standard table 1. Table one should include those important characteristics of the patients you are reporting on: different database sources and sizes, distributions of gender, age, tumor type, etc. It is possible that this paper really has little to do with your cohort other than selecting a few of the many mutations. If that is true, then perhaps de-emphasize figure 1 and do not worry about the table 1 I describe above. However, if you are trying to describe the population impact of the mutations you are characterizing, then we need to have a clearer picture of your cohort.

Response: We are happy to provide further information regarding the cohort of cancer patients studied here. For clarity, the primary candidate driver mutation identification was performed in a cohort comprised of retrospective sequencing data from The Cancer Genome Atlas project and myriad published studies, for which demographic and other cohort-level details have been documented extensively in the literature. To this we added prospective sequencing data acquired in active cancer patients from MSKCC. These patients were not part of a single therapeutic clinical trial but were profiled during the course of their active disease management. The demographic details of this subset of prospectively sequenced patients included a current age of 62 ± 15 years (mean and standard deviation, interquartile range is 18) and 45.2 and 54.8% were male and female, respectively. We have documented the details of our strategy for computational analyses of combined cohorts of both retrospective and prospective clinical data at length in prior work (for example: Chang MT, et al. *Cancer Discov* 2017, Jonsson P, et al. *Nature* 2019, and Gorelick A, et al. *Nature* 2020). Nevertheless, in response we have now added significantly greater detail on the nature of the patient population; its sources and associated citations; and key demographics in the Methods.

Likewise for the mutations that you are considering - I recognize that it is non-trivial to combine mutations from multiple cohorts, but if you are going to present the distributions of AKT mutations in this manuscript as novel data, then we need more details on both the numerator and denominators you are considering. Both are unclear to me. How many mutations were called and exactly how did you arrive at that set? I cannot tell if there was simply a quality filtering of variants, or if there was some additional consideration of driver status based on your computational approach. For example, the lollipop figure 1d shows a lot of grey mutations, including some in grey that are clustered or redundant suggesting potential driver status simply by being recurrent. Were these included in your data for figure 1a? When you say “Candidate driver mutations in AKT1-3 were most common in breast cancers (4.3% of cases)” – does this only refer to your computationally preferred driver mutations, or all mutations that passed sequencing quality parameters.

Response: These are excellent questions and we apologize if there was any confusion or the absence of sufficient details. In total, 1,254 of the 41,075 sequenced tumors harbored *AKT1*, *AKT2*, or *AKT3* somatic mutations (3.1%). These include all *AKT1-3* somatic mutations of any kind that passed primary mutation calling criteria and variant quality thresholds used by the

primary source authors or our own clinical diagnostic laboratory (in the case of MSK-IMPACT data). As with any oncogene in cancer, only a small subset of somatic mutations detected in human tumors are functional driver mutations that lead to aberrant activation and downstream oncogene dependence. We therefore reasoned that reporting only the frequency of presumed driver *AKT1-3* mutations by cancer type would more accurately reflect the subset of each tumor type with cancers presumably driven by their *AKT1-3* mutation than would including all tumors with even randomly acquired *AKT1-3* mutations (either passenger mutations or variants of uncertain significance; those in grey in **Fig. 1d**). In total, 457 of 1,254 unique tumor specimens harbored an *AKT1-3* mutant allele that was considered a driver mutation by one of the three orthogonal methodologies used here. Our group and others have previously observed mutations arising recurrently (rare, but recurrent) for reasons unrelated to selective pressure, but because those alleles are highly mutable by the operative mutational processes in the affected tumor (one reason for what appears to be clusters of seemingly recurrent grey mutations in **Fig. 1d**). This is why recurrence alone is not sufficient to nominate likely drivers when investigating infrequently mutated oncogenes, but more sophisticated methods like those we use here (and have used extensively previously, including in Chang MT, et al. *Nat Biotech* 2016, Chang MT, et al. *Cancer Discov* 2018, among others) are necessary to distinguish between likely passenger and driver mutations. Of course, we cannot exclude the possibility that our computational approaches missed a potential occult driver mutation in these genes that only greater sample numbers would otherwise reveal. We have now added these additional details to an expanded Methods section describing the primary sequencing data and candidate driver mutation detection. We have also clarified the figure legends for **Fig. 1a, b, and d** to ensure the data plotted is more explicitly described.

If you ultimately only characterize a small set of variants, I think we should see the demographics on the tumors with those mutations and a repeat of figure 1a that just considered tumors with driver mutations.

Response: As suggested, we have now added a new **Supplementary Fig. 1a** (see below) showing a corresponding version of **Fig. 1a** that compares the overall representation of disease types among patients with candidate *AKT1-3* mutations and the subset encompassed by the set of mutant alleles we characterized in the current study. Our functional studies profiled a set of mutant alleles that represent the vast majority of all affected cancer types.

I think it is quite reasonable to limit the number of mutations as the authors did. I am not clear the authors tell us specifically what fraction of all mutations are covered by the set of mutations that were generated experimentally. This relates to point #3 above. What disease generated

these mutations, and does this therefore exclude some of the tumor types from being driven by AKT driver mutations?

Response: The mutations tested experimentally here represent 88% of all human tumors harboring candidate *AKT1-3* driver mutations (n = 412 of 467) overall. The spectrum of disease types represented by mutant alleles assessed here were nearly uniformly representative of all affected cancer types overall (see Figure above), including breast cancers, which is the cancer type of greatest frequency of *AKT1-3* mutations (including both missense and indel mutations). We have now updated the manuscript to reflect these results.

I am likewise somewhat unclear as to how to interpret the remaining tail of mutations. Are we to officially brand these all passengers? The authors themselves found at least a small number of unexpected results having made the set of mutations. Is there a recommendation for secondary screening, such as IHC for downstream targets or IHC-appropriate p-AKT antibody? This may be outside of the scope, but I feel this work might at least comment on the path forward for the remaining non-canonical AKT mutants.

Response: This is an excellent question. It is likely premature to label the remaining mutations as passengers. Indeed, our own experience with other mutant oncogenes such as *ERBB2* (Hyman DM, et al. *Nature* 2018), *MEK1* (Gao Y, et al. *Cancer Discov* 2019), and *BRAF* (Yao Z, et al. *Nature* 2017) suggest that rare driver mutations may continue to emerge over time as population-scale sequencing expands (Chang MT, et al. *Cancer Discov* 2018). We agree this deserves mention, so in response we have revised the Discussion of the manuscript to speculate on the emergence of future drivers and alternative ways to screen for these events.

I recognize that this manuscript focuses on mutations, but at least in the introduction could we be reminded whether other mechanisms targeting AKT such as copy number amplification or other abnormality such as gene overexpression plays a role in AKT pathology. Do mutations occur in amplified gene segments such as EGFR or PIK3CA is this more commonly in copy neutral segments such as KRAS.

Response: We are happy to elaborate on mechanisms of AKT activation other than somatic mutations. While focal amplifications of WT *AKT1* have been reported, they are rare. Moreover, exceedingly rare instances of *AKT1-3* fusions have also been reported. Finally, in our prior work on AKT inhibitor sensitivity in *AKT1* E17K mutant patients, we have shown that serial genetic changes can happen at the *AKT1*-mutant locus, but these are quite different from conventional amplification of the mutant such as those in *EGFR* (Hyman DM, et al. *J Clin Oncol* 2017). We have now revised the manuscript accordingly to mention these alternative mechanisms of AKT activation.

Again, while outside the scope of this manuscript, the background might provide some context on epistatic genetic alterations. Specifically, do we know if true AKT driver mutations are found in correlation or anticorrelation with other PIK3CA pathway mutations? Any other known concomitant or co-occurring mutations? Given this cohort, I would think that would be of some importance to assessing the likely efficacy of an AKT inhibitor and further drug development. Specifically, a drug-sensitive AKT mutant patient might still be sabotaged by downstream mutations if they were commonly found together.

Response: This is an excellent point. Our group has in fact published previously on the co-mutational pattern of *AKT1* E17K-mutant tumors (Hyman DM, et al. *J Clin Oncol* 2017). We also appreciate the Reviewer recognizing this topic, while important, is outside the scope of the

current report. We have nevertheless added to the revised manuscript text regarding the importance of the co-mutational pattern of AKT-mutant tumors.

For the indel mutations we would often see the genome alignments for novel mutations presented in the supplement such as an IGV figure. I suppose times have moved on, but when reporting a novel structural oncogenic variant, we would like to see the alignments and the nature of the sequence. This could be important for detection assays going forward.

Response: We are happy to include read alignments for the key indel mutations, which we have added as a new **Supplementary Fig. 1b**.

There is one part of this study that I find truly unorthodox, which is the presentation of a single patient from a clinical trial. Granted this is an exceptional responder, but I think this is really not an appropriate way to present this story. I would exclude that case and report the clinical trial separately with all of the routine clinical trials reporting structures. Trial design. Patient assignment. Doses delivered. Toxicity. Etc. Perhaps that could be co-published with this science? I checked clinicaltrials.gov and this study is still listed as open. I do not think it is generally ethical to report results on a trial that is still ongoing. I would ask the editors for their guidance on this.

Response: The Reviewer raises an important point about our inclusion of an ‘extraordinary responder’ in the context of this manuscript. This patient was treated as part of a pilot study of the ATP-competitive pan-AKT inhibitor AZD5363 in patients with *AKT1*, *AKT2*, or *AKT3* mutations (ClinicalTrials.gov, NCT03310541). The goal of this study was to explore the activity of AZD5363 in two molecularly defined patient populations, those with: 1) *AKT1* E17K-mutant tumors, and 2) patients with non-*AKT1* E17K-mutant tumors. Patients were enrolled to one of three cohorts of up to 12 patients each (breast, prostate, other). Thus, a maximum of 36 patients were to be enrolled. Given the rarity of this patient population, no formal hypothesis testing was pre-specified and this study was therefore considered only a signal-seeking ‘pilot’ study. The primary outcome was objective response according to RECIST version 1.1 (or PCWG3 criteria for patients with prostate cancer). Three or more responses in any cohort of 12 patients was considered promising. However, since this study was initially opened, other studies have more comprehensively reported the activity of ATP-competitive pan-AKT inhibitors in *AKT1* E17K-mutant solid tumors and therefore enrollment of patients harboring this mutation was halted. Overall, a total of 12 patients have been accrued to this protocol since it opened, including six patients with *AKT1* E17K mutations (enrolled prior to closing accrual to these patients) and six patients with non-*AKT1* E17K mutations, including the individual patient reported in this manuscript (see Table below).

Table: Mutations observed in enrolled patients on the AKT study

Mutation	N (%)
AKT1 E17K	6 (50%)
AKT1 D323G	1 (8.3% each)
AKT1 E40K	1
AKT1 L52R	1
AKT2 E17K	1
AKT2 L78_Q79insHANTFVIRCL	1
AKT3 E17K	1

We understand the question about the appropriateness of including a single patient extraordinary responder. Therefore, in response we have decided that it would indeed be more appropriate to report outcomes in all non-*AKT1* E17K patients enrolled in this study (the full

intent-to-treat study population with this overarching biomarker). Moreover, due to slow overall accrual (rarity of the genomic biomarker) we have decided to close this study to further accrual. So, this represents the final study report.

All relevant sections of the manuscript have now been updated including the Methods, which now include expanded details on the study typical of standard clinical trial reporting. The revised Results section now includes a complete accounting of all 6 enrolled patients with non-*AKT1* E17K mutations as do new Supplementary Tables 3 and 4. Briefly, of these six patients, the best overall response included two prolonged partial responses, one stable disease lasting 16 weeks, and three progressive diseases. Partial responses were observed in the prostate cancer patient with the *AKT2* indel as described in the original submission, as well as an endometrial cancer patient with an *AKT1* L52R mutation (see Figure below). This latter response (a durable partial response lasting nearly a year) is another compelling proof-of-principle of our co-clinical trial framework of simultaneously assessing the function and therapeutic sensitivity of key AKT mutant alleles. Another heavily pre-treated breast cancer patient with previously progressive disease enrolled on the basis of an *AKT1* D323G mutation had stabilization of her disease lasting 16 weeks before again progressing. The remaining patients were an *AKT1* E40K-mutant anal adenocarcinoma, *AKT2* E17K-mutant squamous cervix cancer, and an *AKT3* E17K mutant sarcoma, all of whom had progression as their best response. We have now added these additional response details to the manuscript, which help to strengthen our overall findings on the biological and clinical relevance of uncommon driver mutations in AKT1-3. All relevant clinical trial data should now be included in this revision in a manner consistent with Nature publishing guidelines around reporting of clinical trial data. We thank the Reviewer for their advice on more complete data reporting here.

REVIEWER COMMENTS

Reviewer #1 (Remarks to the Author):

The authors have addressed all the queries, concerns and respond to all questions. The authors have improved this new version significantly. The manuscript is ready to be published.

Reviewer #2 (Remarks to the Author):

The authors have added new data and clarified a number of points in the manuscript. There remain a number of points to clarify.

1. Druker and colleagues have demonstrated that BAF3 cells have a mutator phenotype for exogenous plasmids. This is likely due at least in part to the extreme selective pressure that the cells undergo with limiting growth conditions. Have the authors sequences AKT from all BAF3 stable lines to ensure that only the correct version is present. This could greatly influence the results. Indeed, the differences in the mcf10A and the BAF3 data make this a real possibility.

2. The authors have still not explained the induction of PRAS40 and S6 kinase by the F55Y mutant. This needs to be stated more explicitly in the manuscript. A low level of constitutive activity would be predicted to have similar effects on AKT and other substrates that is seen with the other low level kinase molecules. Thus the reason is not clear from the current data and should be stated as such.

3. From the original comments 'Further, the key assay of in vivo growth should be performed as this captures information that is not available in in vitro phosphorylation assays. This would allow an analysis of the sensitivity of several of the mutations to the allosteric and catalytic inhibitors in this study. This would be particularly important for the specific L78_Q79Ins mutation in the illustrative patient.' The authors noted that Covid restrictions prevented the completion of these studies. This remains a major challenge and concern as well as an important weakness in the manuscript. Given the discussions above, it would also be important to include the F55Y mutant.

4. I do agree with the authors on the inclusion of the patient data across the study. The response of a patient with a L52R mutation (mislabeled as L52 in the figure) strengthens the manuscript. However, evidence that this mutation is activating and engenders sensitivity to the inhibitor in both in vitro and in vivo models would strengthen the manuscript.

Reviewer #4 (Remarks to the Author):

The authors have largely addressed the comments of reviewer 3, and the inclusion of the full clinical data set substantially strengthens the manuscript.

The authors should enhance the clarity of the abstract. It was not clear to me on first reading of the abstract that the authors are reporting the results of a clinical trial, in part as “co-clinical trial framework” is ambiguous terminology. The authors should make this more overtly obvious in the abstract.

The authors have partially addressed the prior reviewers comments on Figure 1. Could the authors provide the incidence of AKT mutations by tumor type in supplementary? eg added to Supplementary Figure 1A

For the clinical trial report, a consort diagram should be shown in supplementary.

Figure 5. could be reworded to make it clearer this is the primary report of a clinical trial. Eg Clinical sensitivity of diverse AKT mutations to AKT inhibition in the XXX trial. Start part A with a one sentence description of the trial.

Figure 5C The authors state that a CT and PET scan is shown, but it looks like a CT and bone scan instead. Suggest that authors check this.

Sup figure 10. Suggest extend the supplementary figure legend to describe the clinical data set and results more fully. Response rates, median PFS with confidence intervals. I acknowledge the data set is small, but provision of these basic figures is important.

** See Nature Research's author and referees' website at www.nature.com/authors for information about policies, services and author benefits.

Reviewer #1 commented that we “*have addressed all the queries, concerns and respond to all questions.*” They also mention that “*the authors have improved this new version significantly. The manuscript is ready to be published.*”

Reviewer #2 commented that we “have added new data and clarified a number of points in the manuscript.” Nevertheless, they also had a few remaining points for clarification, which we address below:

Druker and colleagues have demonstrated that BAF3 cells have a mutator phenotype for exogenous plasmids. This is likely due at least in part to the extreme selective pressure that the cells undergo with limiting growth conditions. Have the authors sequenced AKT from all BAF3 stable lines to ensure that only the correct version is present. This could greatly influence the results. Indeed, the differences in the mcf10A and the BAF3 data make this a real possibility.

Response: We appreciate the Reviewer’s insight regarding the potential obstacles with the BaF3 transformation assay. Druker and colleagues found artifacts in transformation assays in BaF3 cells arising due to additional mutations gained in the transgene while following a workflow that consisted of culturing BaF/3 cells expressing the transgene in IL-3-deficient medium to initiate a cytokine-independent growth assay lasting approximately three weeks. We agree that selecting out only those BaF3 cells that outgrow these limiting and restrictive conditions and performing subsequent analyses only with this subpopulation of cells increases the likelihood of drawing incorrect conclusions regarding the transgene being expressed. However, this does not apply to the experiment we performed. Our experimental workflow for BaF/3 cells was fundamentally different. We did not select a population of stable Ba/F3 cells capable of growing in IL-3-deficient media for our pathway analysis or successive experiments. We cultured all BaF3 stable lines in the presence of 10ng/ml IL-3 at all times, and the cells we used for our experiments were never subjected to IL-3 withdrawal prior to performing the experiments. There was no selection bias for IL-3 independence. After generating stable BaF3 lines post-transduction and antibiotic treatment, we verified the expression of the desired transgene by immunoblotting. Cells from these stable lines were washed, seeded into 6-well plates, and were assessed for pathway activation after only an overnight incubation with IL-3-deficient media. Likewise, for the IL-3-independent transformation assay, we used the BaF3 cells stably expressing the desired constructs, washed them to remove IL-3 from the media, and incubated them in IL-3-free media for the duration of study period (total of 7 days), all of which is described in detail in the Methods Section. Consequently, we have not subjected these cells to the extreme selective pressure of limiting growth conditions and certainly not for any extended period of time, as is applicable for the findings of Druker and colleagues.

In addition to having performed a very different experiment than Druker and colleagues, we have also included experimental controls that facilitate correct inference from data. For both pathway analyses and IL-3-independent growth assay experiments, we have included BaF3 cells stably expressing GFP and WT AKT. We also tested five other AKT1 variants (R15Q, D44N, R48H, R370C, E464K) predicted to be non-activating based on both our computational analysis and experimental data with MCF10a cells. None of these negative controls and likely passenger variants were able to spontaneously transform BaF3 cells. All of these cells were also subjected to identical growth and test conditions as cells expressing other activating mutants. The only mutants that transformed Ba/F3 cells to IL-3-independence were the ones also capable of causing growth factor-independent pathway activation in both MCF10a and BaF3 cells.

Collectively, the experimental design and inclusion of appropriate controls effectively exclude the possibility that unwanted mutations evolved in the constructs in AKT1/2 and led to the transformed Ba/F3 phenotype we observed. It is therefore reasonable to conclude that transformation of Ba/F3 cells in our assays is due to the degree of pathway activation conferred by the tested activating and oncogenic AKT1/2 mutants.

The authors have still not explained the induction of PRAS40 and S6 kinase by the F55Y mutant. This needs to be stated more explicitly in the manuscript. A low level of constitutive activity would be predicted to have similar effects on AKT and other substrates that is seen with the other low level kinase molecules. Thus the reason is not clear from the current data and should be stated as such.

Response: We thank the reviewer for this suggestion and have now revised the main text of the manuscript to clearly state this finding.

I do agree with the authors on the inclusion of the patient data across the study. The response of a patient with a L52R mutation (misabeled as L52 in the figure) strengthens the manuscript. However, evidence that this mutation is activating and engenders sensitivity to the inhibitor in both in vitro and in vivo models would strengthen the manuscript.

Response: We thank the reviewer for the comment on the L52R-mutant patient. For the purposes of clarification, we note that the manuscript already includes detailed results regarding the impact of the L52R mutation on pathway activation (**Figs. 2a** and **2d**), IL-3 independence (**Fig. 2e**), and AKT inhibitor sensitivity (**Fig. 4a-b**) in all of our *in vitro* assays with both MCF10a and Ba/F3 cells. We apologize if this was not clear in the manuscript and have now relabeled the L52R-mutant patient in **Fig. 5b** to ensure greater clarity.

Reviewer #4 mentioned that we “*have largely addressed the comments of reviewer 3, and the inclusion of the full clinical data set substantially strengthens the manuscript.*” They had a few additional queries, which we address below:

The authors should enhance the clarity of the abstract. It was not clear to me on first reading of the abstract that the authors are reporting the results of a clinical trial, in part as “co-clinical trial framework” is ambiguous terminology. The authors should make this more overtly obvious in the abstract.

Response: We agree and in response we have now revised the abstract with the following language to clarify we are reporting the results of a clinical trial: "We conducted a phase II clinical trial testing the AKT inhibitor, AZD5363, in patients with solid tumors harboring AKT alterations (NCT003310541). Twelve patients were enrolled including 6 with AKT1-3 non-E17K mutations. As proof-of-concept, a patient with an AKT2 duplication had a prolonged radiographic and clinical response. The full clinical trial results are presented here."

The authors have partially addressed the prior reviewers' comments on Figure 1. Could the authors provide the incidence of AKT mutations by tumor type in supplementary? eg added to Supplementary Figure 1A

Response: We have now added the incidence of AKT1-3 mutations by tumor type to **Supplementary Fig. 1A** for reference and as reflected graphically in **Fig. 1**.

For the clinical trial report, a consort diagram should be shown in supplementary.

Response: We have now added a consort diagram as **Supplementary Fig. 10a**.

Figure 5. could be reworded to make it clearer this is the primary report of a clinical trial. Eg Clinical sensitivity of diverse AKT mutations to AKT inhibition in the XXX trial. Start part A with a one sentence description of the trial.

Response: We have now rewritten both the Figure title and panel A text to more clearly state these results are the primary reporting of the associated clinical trial.

Figure 5C The authors state that a CT and PET scan is shown, but it looks like a CT and bone scan instead. Suggest that authors check this.

Response: We thank the Reviewer for their keen eye, and we have now corrected the legend to refer to CT and bone scans.

Sup figure 10. Suggest extending the supplementary figure legend to describe the clinical data set and results more fully. Response rates, median PFS with confidence intervals. I acknowledge the data set is small, but provision of these basic figures is important.

Response: We have now revised both the legend of **Supplementary Fig. 10** as well as the main text to ensure these results are more fully communicated.

REVIEWERS' COMMENTS

Reviewer #2 (Remarks to the Author):

Sequencing AKT in the clones studied particularly in the stable lines is good practice. Not only would this deal with the potential mutator phenotype but is the only way to validate the reagents used to establish that the lines were not mixed up along the way. With diverse cell lines this is done by fingerprinting but with isogenic lines such as used here, this is only way to adequately validate the reagents used.

Reviewer #4 (Remarks to the Author):

The authors have addressed my comments.

Reviewer #2 (Remarks to the Author):

Sequencing AKT in the clones studied particularly in the stable lines is good practice. Not only would this deal with the potential mutator phenotype but is the only way to validate the reagents used to establish that the lines were not mixed up along the way. With diverse cell lines this is done by fingerprinting but with isogenic lines such as used here, this is only way to adequately validate the reagents used.

We thank the reviewer for their input. We have performed Sanger sequencing of the exogenous AKT1 from the stable Ba/F3 cells we generated, and are happy to report that, aside from the intended mutation in each stable line, no other unwanted mutation was identified in any of the lines. This rules out the possibility that our results from our assays were in any way influenced by undesirable mutations acquired in the transgene.

We are outlining here reports from almost all activating AKT1 alleles that induced Ba/F3 transformation, as well as a couple of alleles that produced no effect on pathway activation or transformation of Ba/F3 cells.

1. AKT1 R15Q

CGA > CAA

2. AKT1 E17K

GAG > AAG

3. AKT1 E40K

GAG > AAG

4. AKT1 D44N

GAT > AAT

5. AKT1 L52R

CTC > CGA

6. AKT1 F55Y

TTC > TAT

7. AKT1 I65-I75dup

ACGGAGCGGCCCCGGCCCAACACCTTCATCATC > ACGGAGCGGCCCCGGCCCAACACCTTCATCATC
ACGGAGCGGCCCCGGCCCAACACCTTCATCATC

8. AKT1 E66-Q79dup

GAGCGGCCCCGGCCCAACACCTTCATCATCCGCTGCCTGCAG > GAGCGGCCCCGGCCCAACACCTTCATCATCCGCTGCCTGCAG
GAGCGGCCCCGGCCCAACACCTTCATCATCCGCTGCCTGCAG

9. AKT1 P68-C77dup

CCCCGGCCCAACACCTTCATCATCCGCTGC > CCCC GGCCCAACACCTTCATCATCCGCTGC
CCCCGGCCCAACACCTTCATCATCCGCTGC

Reviewer #4 (Remarks to the Author):

The authors have addressed my comments.